# Genomic diversity and structure of prehistoric alpine individuals from the Tyrolean Iceman's territory

Myriam Croze [1,2,5], Alice Paladin [1], Stefania Zingale [1], Sofia Alemanno[1], Franco Nicolis[3], Elisabetta Mottes [3], Frank Maixner [1], Annaluisa Pedrotti[4], Torsten Günther [2], Albert Zink [1] & Valentina Coia [1] ✉

The Eastern Italian Alps played a crucial bridging role between Mediterranean and Northern alpine populations since Prehistory. However, few prehistoric individuals from that region have been genomically analysed so far. Among them, the Iceman (Copper Age, 3368-3108 BC) showed a relatively high Anatolian-Neolithic-related ancestry and low Hunter-Gatherers (HGs)-related ancestry. To investigate how the genomic structure of alpine groups varied over time and to contextualize the Iceman, we analysed 47 alpine individuals dated from the Mesolithic (6380-6107 BC) to Middle Bronze Age (1601-1295 BC). The Mesolithic genome reveals genetic admixture between Western and Eastern HGs that occurred from ~13700 − 8300 BC. Most individuals from the Neolithic onwards present a genomic structure resembling that of the Iceman, supporting genetic continuity. Few individuals carry different ancestries, such as the Steppe-related ones appearing ~2400 BC. Finally, the study suggests local and non-local admixture events between HGs and Neolithic farmers from this alpine area.

The Eastern Italian Alps, including the Trentino-Alto Adige region (EIAlp) considered in this study, are an important area for expanding our knowledge of Prehistory and the major demographic events that occurred in Europe and contributed to shaping the gene pool of today's European populations[1].

The first event is associated with the Hunter Gatherers (HGs), including two major groups: Western Hunter-Gatherers (WHGs) related to the Villabruna cluster[2] and the Eastern Hunter-Gatherers (EHGs) from present-day Russia and Ukraine[3]. The second regards the migrations of Early Neolithic (EN) farmers from Anatolia (from ~7000 BC)[1,4,5] who admixed with local HGs groups, to varying degrees depending on the area[1,6]. The last one is instead related to the expansion of Pontic-Caspian Steppe pastoralists (from ~3000 BC)[7,8].

Indeed, due to the presence of wide valleys (e.g. Adige Valley) and mountain passes (e.g. Passo del Brennero/Brennerpass), the EIAlp played a crucial role as a bridge between populations from northern and southern alpine territories, favouring cultural exchanges and contacts. However, the mountain environment may also have influenced such exchanges and interactions among the alpine groups.

The early population history of Europe was heavily influenced by the Last Glacial Maximum (LGM, 30,000–16,500 years ago) when the Alps were covered by thick glaciers and continental Europe was accessible from Italy, mainly from the Balkan Peninsula, and bridges emerging from the Adriatic Sea[9]. When the large Pleistocene glaciers retreated (~18,000 years ago), the EIAlp was entered by HG groups of the final phase of the Upper Palaeolithic, who possibly arrived from a

[1]Institute for Mummy Studies, Eurac Research, Viale Druso 1, 39100 Bolzano, Italy. [2]Department of Organismal Biology, Evolutionary Biology Centre, Uppsala University, SE- 752 36 Uppsala, Sweden. [3]Provincia autonoma di Trento, UMSt Soprintendenza per i beni e le attività culturali, Ufficio beni archeologici, Via Mantova 67, 38122 Trento, Italy. [4]Department of Humanities, University of Trento, Via T. Gar, 14, 38122 Trento, Italy. [5]Present address: The International ImMunoGeneTics Information System (IMGT), Institute of Human Genetics (IGH) UMR9002, CNRS, Univ. Montpellier (UM), Montpellier, France. ✉e-mail: valentina.coia@eurac.edu

more southern area (e.g. Po Plain region)[10,11]. During the temperate climate phase (Late Glacial interstage; ~14,500 and 12,500 years ago) numerous sites (e.g., open-air sites, rock shelters, caves) referable to the Epigravettian culture, were attested (e.g., Arco via Serafini, Riparo Dalmeri, Riparo Villabruna)[12,13] and with the beginning of the Holocene, temperate climatic conditions favoured the settlement of Early (~9000 BC, Sauveterrian) and Late Mesolithic (ME) (~6500 BC, Castelnovian) HGs groups on the valley floors (Trentino area) with subsequent expansion into higher altitudes[14–16].

In the EIAlp, transition from a ME subsistence economy (based on hunting and gathering) to a new Neolithic (NE) productive one (based on agriculture and animal husbandry) is attested between 5100 and 4800 BC (Trentino area)[17,18].

Even if in Europe EN farmers migrations followed complex and multifaced dynamics in the various geographical areas[19], in general, two major routes have been described[20–23]. One along the Mediterranean coastline, associated with the Impressed Ware culture, and another through the Balkans, which led to the formation of the Linearbandkeramik (LBK) culture in central Europe and partly contributed to the formation of the first NE groups in northeastern Italy[19]. In the latter area, including the EIAlp and the Po Valley south of the Alps, both NE economic and cultural innovations took root, resulting in various Padano-Alpine cultural groups (e.g. Fiorano, Vhó, Isolino, Gaban)[19,24]. The potteries of this period also show influences from the Impressed Ware ceramic group of central Italy[25,26]. From ~4800 BC the cultural traits of the "Gaban group", which was named after the archaeological site (Trentino area), began to be gradually replaced by the diffusion of the Square-Mouthed Pottery Culture (Vasi a Bocca Quadrata, VBQ), distributed mostly in northern and central (Toscana) Italy[19]. From the mid-4th mill. BC, within the Copper Age (CA; ~3500–2200 BC), significant social changes occurred in the EIAlp with the formation of cultural groups distinguished mainly by different funerary practices such as burials in simple earthen pits with bodies placed on the left side (like Remedello group)[27] as well as supine inhumations (Spilamberto group)[28] and human remains manipulation and cremation (Grotticelle group)[29].

During the Late CA (~2700–2400 BC) and the transition phase from the CA to the Early Bronze Age (EBA, ~2400–2200 BC), the exploitation of copper ores began in the Trentino-Alto Adige region[30,31], continuing in the EBA with the Polada culture[32,33].

In the Adige valley, contacts between groups from both sides of the Alps (from Northern–Central Europe, transalpine Middle Danubian area and the Po River Valley) are clearly documented by the discovery, in this valley, of numerous objects (e.g. Baltic amber beads) and weapons referable to those areas[33–35].

Despite the wealth of archaeological information on the different cultures present in the EIAlp during Prehistory, very little is known about the genomic diversity and structure of alpine prehistoric individuals from this and surrounding alpine areas. In fact, only a few prehistoric individuals had been genomically analysed before this study. These involved the Late Upper Palaeolithic individual of Villabruna (~14,000 BP; Veneto region) that gives the name to the Villabruna genetic cluster[2], and three CA individuals, including the Tyrolean Iceman, later referred to only as Iceman[36–38]. Furthermore, palaeogenomic data on prehistoric specimens from present-day Italy are still limited and only concern specimens recovered south of the Alps[7,39–47].

A recent study based on a high coverage genome of the Iceman (3368–3108 BC)[37], found that this individual carried a high percentage (90% ± 2.5%) of ancestry associated with EN farmers from Anatolia and low WHGs-related ancestry. The EN farmer-related ancestry has been found to be the highest among contemporaneous European individuals. This may indicate rather isolated alpine groups with limited gene flow from HG-ancestry-related populations or smaller population size of HG groups in this region during the 5th and 4th millennia BC[37]. Additionally, the study reported further information on some phenotypic

traits of the Iceman, in addition to those previously described[36]. However, although the study provides important initial insights into the genetic history of prehistoric EIAlp, it is based only on a single individual, and its findings and interpretation need to be supported by genomic data from more specimens from this alpine territory.

Studies carried out in north-eastern Italy on CA and Bronze Age (BA) individuals[7,41,42], show genomic affinity of CA individuals to EN farmers, similarly to the Iceman. In addition, these studies suggest that the genetic component related to the herders from the Steppe (Steppe-related ancestry) arrived in northern Italy at least by ~2000 BC[42]. Furthermore, no genomic data are available on individuals from the ME or NE periods from northern Italy, apart from one early ME individual (north-western Italy) for whom only mitochondrial DNA and low-resolution data are available[46].

We successfully generated new paleogenomic data (shotgun and capture) from 47 prehistoric/protohistoric (hereafter referred to as prehistoric) alpine individuals from 17 archaeological sites from the EIAlp. Additionally, we report 34 new radiocarbon ($^{14}$C) dates that, besides those already available, have resulted in a dataset that includes individuals from the Late ME (6380–6107 BC) to the Middle Bronze Age (MBA, 1601–1295 BC) (Fig. 1 and Supplementary Data 1). The data were used to understand how the genomic structure of alpine prehistoric individuals varies over time. This allows us to investigate the genetic impact related to migration events during the ME and NE, and the extent of admixture between NE farmers and local HGs, as well as the possible impact and timing of the arrival of Steppe-related ancestry in this area. Furthermore, by extending the genomic analyses to a larger number of specimens contemporary to the Iceman, we aim to offer a more comprehensive picture of the genomic structure of prehistoric alpine individuals during CA. Finally, we intend to provide an initial insight into the possible social practices of past alpine groups and give information on some phenotypic traits that have also been investigated in the Iceman.

In this work, we reveal that the ME alpine individual shows a genetic admixture between Western and Eastern HGs that occurred between 13,700 and 8300 BC. Furthermore, we found a substantial genetic shift from ME to NE and genetic continuity from NE (from at least ~4600–4400 BC) onwards, with a low and constant contribution from HGs. In addition, we report the estimated time of admixture between NE Anatolian farmers and ME (~6100–5100 BC), supporting both local and non-local admixture in the EIAlp. Our study also finds a low genetic impact related to the migration of herders from the Steppe on alpine groups, although it suggests that Steppe-related ancestry appeared in EIAlp earlier than in other Italian regions, including northern Italy. Finally, we found that the Iceman shows the same ancestral genomic model and similarity in phenotypic traits as the other individuals sampled from the CA and most of the other alpine individuals analysed, confirming and extending previous results. However, the Iceman differs from all other alpine individuals analysed in its maternal and paternal lineages, suggesting a slightly different genetic history.

## Results
### Alpine dataset
Paired-end genomic libraries of DNA extracts from 52 samples (Pars petrosa or teeth) were tested for their content of endogenous human DNA (HR) and authenticity using shotgun sequencing (Supplementary Data 2; Supplementary Information Text S3). The samples with HR > 1% were further enriched for 1240 K SNPs across the human genome, as well as for the Y-Chromosome and mitochondrial DNA (mtDNA) (Supplementary Data 3). Individuals ($n = 5$) who did not fulfil the quality criteria were excluded from the study (Methods and Supplementary Information Text S3). The final successfully analysed alpine genomes (45 capture and 2 shotgun), with low contamination and typical ancient DNA (aDNA) damage patterns, had a mean coverage

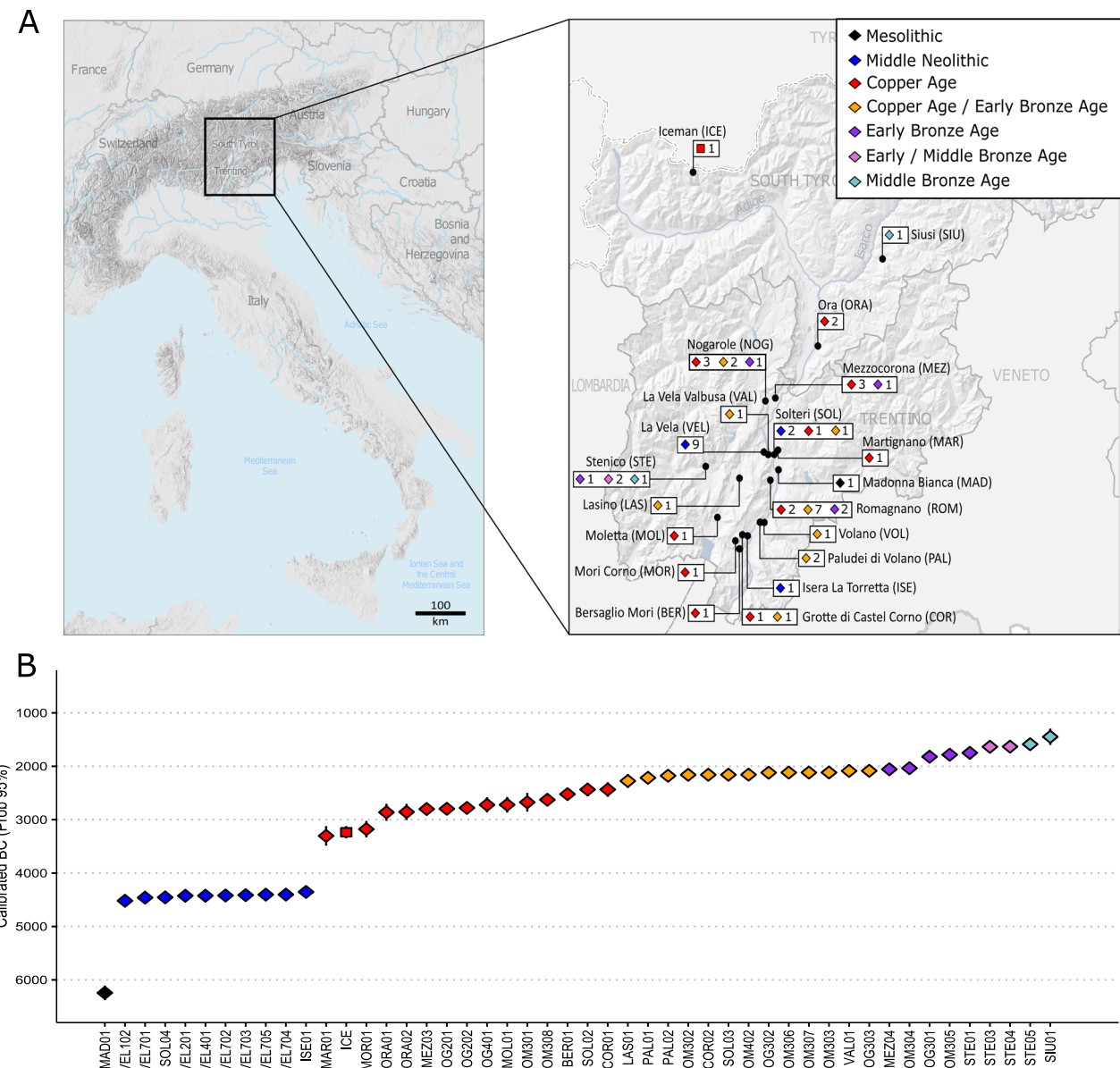

**Fig. 1 | Geographical distribution of sampling sites and radiocarbon dates.**
**A** Geographic map of Italy and the Alpine arc (left) was performed by using ESRI's ArcGIS Pro software, and a zoom-in (right) of the Trentino Alto-Adige region in the EIAlp with the location of the archaeological sites from this study (Iceman[37] and Ora/Auer[38]). The number of sampled individuals from each site is near the symbol, coloured according to chronology. The first three letters of the site, also used in the sample identification number, are given in brackets. **B** Mean radiocarbon dates and calibrated intervals (cal. 2-sigma, 95%) for each dated individual (more details in Supplementary Data 1 and Supplementary Information Fig. S16).

from 0.012× to 0.587× while the number of SNPs on the 1240 K panel[48] ranged from 33,981 to 1,046,042 SNPs sites (Supplementary Data 4). Available data from the other three alpine CA individuals (Iceman and individuals from Ora[37,38]) were included reaching a dataset of 50 individuals from the EIAlp which comprises: one Late ME, eight Middle NE (MN), sixteen CA, sixteen CA/EBA and nine from the EBA to the MBA (Supplementary Data 1 and Supplementary Data 4). The dataset includes a similar number of females and males, including several subadults (26 XX and 24 XY; Supplementary Data 4). Among the sixteen related individuals (based on full consistency between at least two of the applied methods; Supplementary Data 5; Supplementary Information Text S4), only the sample of the related pair, which showed the highest number of SNPs, was kept for population genetic analysis, yielding a final dataset of 41 unrelated alpine individuals (Supplementary Data 4 and 5).

## Admixture events between WHGs and EHGs in the ME genome from the EIAlp

The Late ME male alpine individual (MAD01; 6380–6107 BC) differs from all other most recent alpine individuals in the Principal Component (PCA) and clustering analyses (Fig. 2A, and Supplementary Information Fig. S22). It shows a strong genomic affinity to WHGs and the Upper Palaeolithic individual from Villabruna, which gave its name to the Villabruna genetic cluster comprising most of the HGs from western Eurasia analysed so far[2]. Furthermore, a multidimensional scaling plot (MDS) on pairwise outgroup $f_3$-statistics performed using the dataset of Posth and colleagues[44], confirms that MAD01 falls together with WHG groups (Villabruna and Oberkassel clusters). Additionally, MAD01 shifts in the plot towards EHGs individuals from present-day Russia (Sidelkino) (Fig. 2B and Supplementary Data 11 and 12).

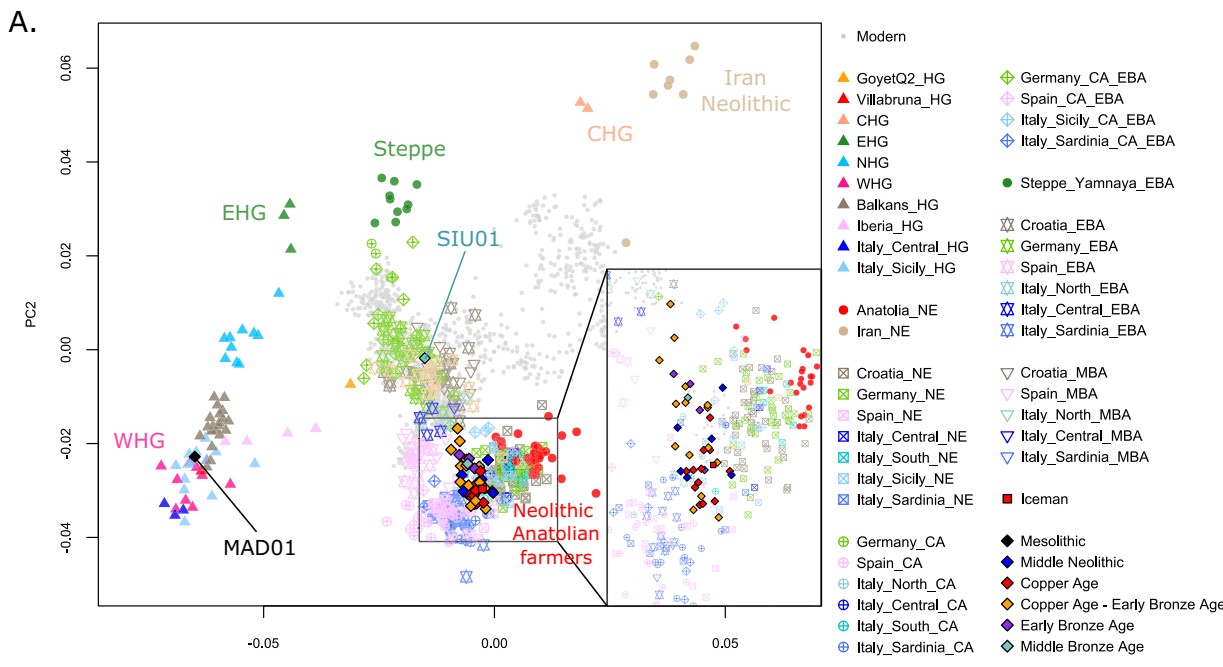

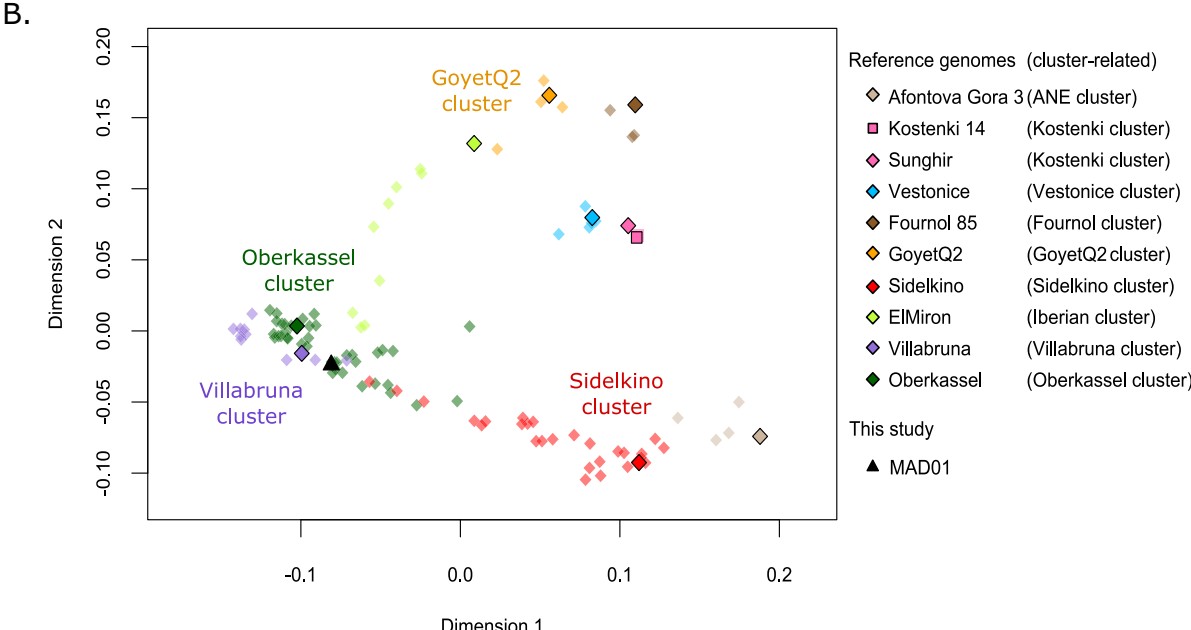

**Fig. 2 | Principal components analysis (PCA) and multi-dimensional scaling (MDS) representations. A** PCA of ancient individuals from this study projected onto the variation of present-day individuals from Western Eurasia (refer to Supplementary Data 10). **B** MDS plot of Eurasian HGs individuals based on $1 - f_3$ (Mbuti, pop1, pop2) statistic. Comparative data from ref. 35. ANE Ancient North Eurasian (refer to Supplementary Data 11).

Our qpAdm analysis confirms that MAD01 can be best modelled (p-value = 0.145) as the admixture between the two sources of Villabruna (83.6 ± 3.7%) and EHGs-Sidelkino (16.4 ± 3.7%) rather than Oberkassel and EHGs although this model is also possible (p-value = 0.0335) (Supplementary Data 16). We then ran qpAdm using Villabruna and Fournol as sources (the latter ancestry is absent in Villabruna but not in Oberkassel), however the model was rejected (p-value = 0.00118) (Supplementary Data 16). The time of the admixture event in MAD01 between a group of WHGs and EHGs estimated by DATES ranged between 13735 BC and 8326 BC (Supplementary Information Text S7 and Fig. S26, Supplementary Data 20).

The analyses of the unilinear markers in this individual also showed that MAD01 carries the Y-Chromosomal I*(I2a1b1a2b) and the

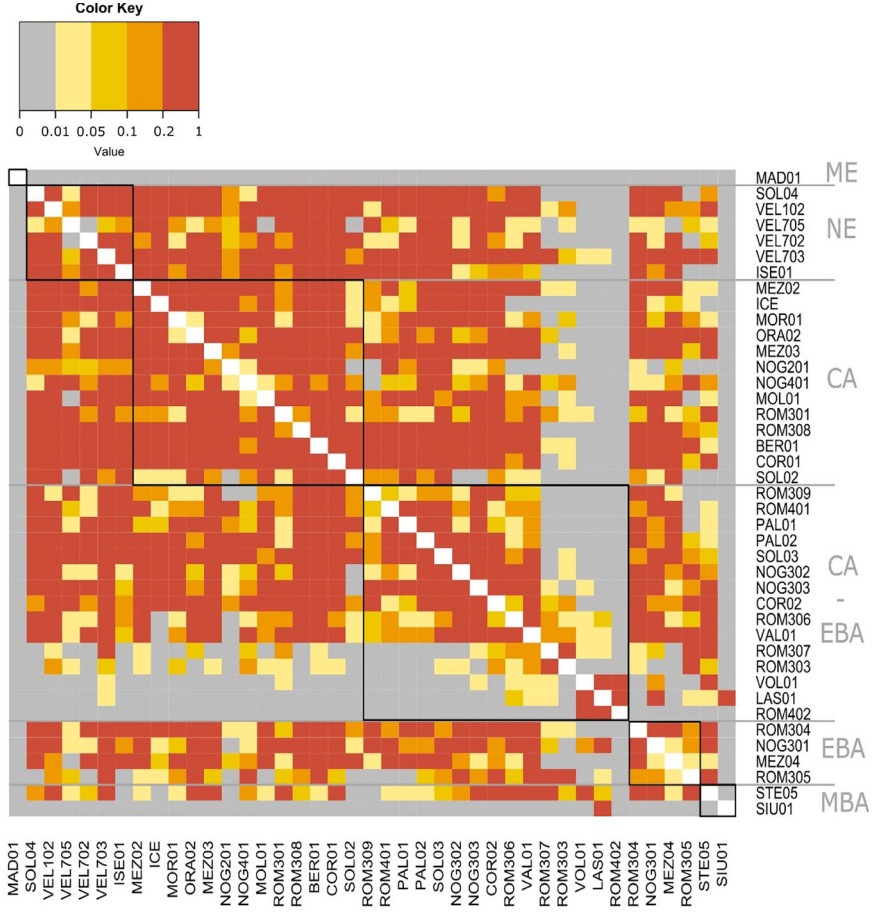

**Fig. 3 | Pairwise-qpWave analysis among prehistoric alpine individuals.** The black squares indicate individuals from the same chronology (NE Neolithic, CA Copper Age, CA-EBA Copper Age-Early Bronze Age, EBA Early Bronze Age, MBA Middle Bronze Age).

mtDNA U5* (U5b2b3) haplogroups, typically found in individuals with WHGs-related ancestry[2,44].

### High early anatolian farmers-related ancestry with few changes in the alpine genomes over time

The PCA plot (Fig. 2A, Supplementary Information Fig. S19A and Supplementary Data 10), shows that the majority of alpine individuals from MN to MBA have genomic affinity with each other and mostly with present-day Sardinians. In line with these results, pairwise-qpWave analysis revealed that most of our sampled individuals (excluding seven: ROM303, ROM307, ROM402, NOG302, LAS01, VOL01, and SIU01), share similar ancestries and can be grouped together (Fig. 3). Furthermore, in the PCA graph, most alpine genomes shift toward those from the EN farmers of Anatolia, with whom they share a similar genetic pattern. Specifically, the alpine individuals are in an intermediate position (zoom Fig. 2A and Supplementary Information Fig. S20A) between the NE individuals of Central and Balkan Europe and those of Spain, and the $f_3$ and $f_4$-statistics did not reveal specific affinities with any of these groups (Supplementary Information Text S6, Figs. S27 and S28).

When we formally tested the admixture models by qpAdm analyses (Fig. 4 and Supplementary Data 17), we found that most alpine individuals from MN to MBA (excluding the seven ones that deviate the most from the general pattern, as detailed in the subsequent paragraph), can be best modelled by a two-way model of admixture with high EN-Anatolian farmers and low HGs-related ancestry. Indeed, the EN-related ancestry ranged from 90.9 ± 2% (COR02, dated to CA/EBA, 2276–2041 BC) to 80.3 ± 2.2% (ROM401, relative dating corresponds to CA/EBA, no $^{14}$C data available) while local HG-related ancestry (MAD01)

varied in these two individuals from 9.1 ± 2% to 19.7 ± 2.2%, respectively (Fig. 4 and Supplementary Data 17).

As a further step, in order to test whether the HGs-related component found in alpine MN individuals could be related to different groups of HGs rather than MAD01, an $f_4$-statistical analysis on the form $f_4$ (HG_test, MAD01; MN, Mbuti) was performed with different sources of proxy HGs (Supplementary Data 15). Most $f_4$-statistics were negative (except for Vela_Spila from Croatia and Villabruna), but no test was significant (z-score between −2.8 and +1.069), indicating that no other HGs used for the analysis fit better than MAD01 as a source of HGs ancestry in alpine MN. Notably, the two tests with the strongest affinities with MAD01 (Zscore < −2.7) were the comparisons with ME from Iron Gates, practically excluding the Balkans as a region where NE individuals could have obtained substantial amounts of their HG ancestry before reaching the EIAlp (Supplementary Data 15).

The CA Iceman is after COR02 (CA/EBA from Grotte di Castelcorno, Trentino, Supplementary Information Text S1), the second alpine individual carrying the highest (90.5 ± 2%) and lowest (9.5 ± 2%) proportions of EN Anatolian and HGs-related ancestry, respectively. These values are very similar to those that were found in the previous study by Wang and collaborators[37] (89.5 ± 2.5% and 10.5 ± 2.5%).

We additionally found that the proportions of HG-related ancestry in the alpine individuals from the different chronologies differ only slightly (Fig. 4 and Supplementary Data 17). Comparable percentages of HGs-related ancestry (MAD01) were also observed when qpAdm analyses were performed, grouping the alpine individuals based on their chronologies and taking into account differences in their ancestral components (Fig. 4, Supplementary Information Fig. S24 and Supplementary Data 19). A similar pattern

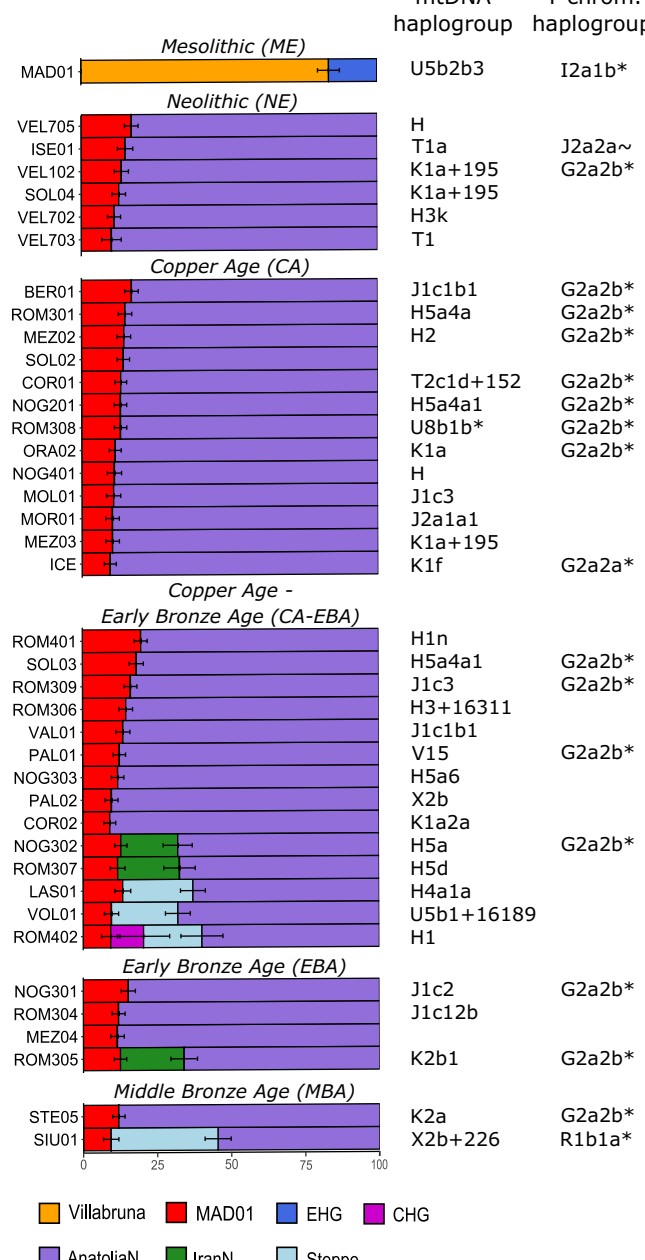

**Fig. 4 | qpAdm analysis and admixture models of each prehistoric alpine individual.** Each colour represents one of the seven sources used in the models. Number of individuals = *n*. Villabruna (*n* = 1), MAD01 (Madonna Bianca, *n* = 1), EHGs (Eastern Hunter Gatherers, *n* = 18), CHG (Caucasus Hunter Gatherers, *n* = 2), AnatoliaN (Anatolian Neolithic farmers, *n* = 22), IranN (Iranian Neolithic farmers, *n* = 4) and Steppe (Yamnaya, *n* = 10). The standard error for each source proportion is indicated by the error bars. No significant model could be found for one individual (ROM303). Haplogroups of the mitochondrial DNA (mtDNA) in all individuals and Y-Chromosome haplogroups in males are indicated on the right-hand side of the graph.

with few changes of the HG-related component over time can be observed in Sardinia and Sicily, as suggested by our qpAdm analysis on other available prehistoric individuals from Italy (Supplementary Data 19). Focusing on the CA period in the alpine region and northern Italy (samples from Remedello and Broion), our qpAdm analysis additionally shows similar (low) proportions of HG-related ancestry (MAD01) in these regions (Supplementary Data 19 and Supplementary Information Fig. S25).

The time of the admixture events between WHG and Anatolian farmers in EIAlp groups or individuals was estimated with DATES (Supplementary Data 20), and the ancestral models with the two source populations used for the analysis were verified with qpAdm analysis (Supplementary Data 19 and Supplementary Information Text S7). The admixture time estimated in the alpine MNs ranged from 6073 to 5117 BC (Supplementary Information Fig. S26).

### Different ancestries occasionally detected in the EIAlp from the CA-EBA, and a low genetic impact related to Steppe herders

Most of the alpine sampled individuals, across the various chronologies, display a similar genomic structure and ancestry (Figs. 3 and 4). However, seven of them deviate from this general pattern (Supplementary Information Fig. S23). The three CA-EBA (LAS01, VOL01, and ROM402), which shift in the PCA plot in the direction of genomic diversity of BA individuals with high Steppe-related ancestry (Yamnaya), and one MBA individual (SIU01), which behaves as an outlier in the plot (Fig. 2A). The latter cluster with BA individuals from central Europe and present-day western European populations (Supplementary Information Fig. S19A, Fig. S20C and Fig. S20D).

Overall, *f*4-statistics do not show clear genetic similarities between alpine CA-EBAs and specific groups (Supplementary Data 14 and 15; Supplementary Information Text S6 and Fig. S27). Using qpAdm, three-way best-fitting model (with HG-MAD01, EN from Anatolia and herders from the Steppe as sources) for the individual LAS01 (2402–2149 BC), VOL01 (relative dating corresponds to CA/EBA, no [14]C data available) and for SIU01 (1601–1295 BC) was found. Instead, a four-way admixture model, with the addition of one component from Caucasian HGs (CHG; 11.1%), was found in one individual (ROM402) (2272–2041 BC) (Fig. 4 and Supplementary Information Fig. S24 and Supplementary Data 17 and 18). This individual had possibly a higher level of CHG-related ancestry than that included in the Steppe-related ancestry present in the genome of the population used as a source in the model (Steppe-herders).

The percentage of the Steppe-related ancestry in the EIAlp ranged from 19.7 ± 7.1% (ROM402) to a maximum of 36.1 ± 2.6% in SIU01, which explains the position of this outlier in the PCA plot (Fig. 2A). The Steppe-related component value found in SIU01 is higher than those estimated in BA individuals from Broion in northern Italy (29.5 ± 3.8) with a comparable age (1615–1431 BC) (Supplementary Data 19). Based on our qpAdm analyses, which include the ancient data available from Italy to date, this component was first identified in our CA-EBA individual LAS01 dated 2402–2149 BC, while the oldest individual (Bell-Beaker culture) carrying the Steppe-related component in northern Italy is dated to 2200–1930 BC[41,49] (Supplementary Information Fig. S25, and Supplementary Data 19). The admixture time between local CA and Steppe-pastoralist-related ancestry (Yamnaya) in LAS01 estimated by DATES ranged between 2912 and 2219 BC (Supplementary Information Text S7 and Fig. S26, Supplementary Data 20).

The other three alpine individuals that differ from the general pattern include NOG302 (CA-EBA, 2202–2037 BC), ROM307 (CA-EBA, 2200–2035 BC), and ROM305 (EBA, 1878–1688 BC). Unlike the other four individuals, they show the most parsimonious and best fitting model (*p*-value between 0.066 and 0.224) when including the Iranian Neolithic (IN) as a source in addition to EN-Anatolia and HGs (Fig. 4 and Supplementary Data 17) with the IN-related ancestry ranged from 19.2 ± 4.9% (NOG302) to 21.5 ± 4.5% (ROM305). In fact, when running qpAdm on these three individuals with a model that includes the Steppe-related ancestry as a source instead of the IN-related ancestry, the three-way model either does not fit (*p*-value < 0.01) (NOG302 and ROM305) or fits with a lower *p*-value (0.012, ROM307). Additionally, based on our comparative qpAdm analysis (Supplementary Data 19), the IN-related component was not detected in any other prehistoric (CA and BA) individuals from northern Italy. On the other hand, it has been found in NE individuals from southern Italy who show this

ancestry in addition to those related to HGs and EN farmers. Moreover, central Italian NE and CA individuals best fit with a two-way model (HG and EN), but also a three-way model is possible, which includes the IN-related ancestry in agreement with the analyses performed by[40] (Supplementary Data 19).

## Unilinear markers, kinship and phenotypic traits

In agreement with autosomal data, Y-chromosome haplogroup analysis shows very few changes in the EIAlp from MN to MBA, with the males analysed carrying the same haplogroup G2a2* associated with the early NE settlement of Europe[48]. Only two individuals, besides the ME one, had different haplogroups (ISE01, J2a2a and SIU01, R1b-P312) (Fig. 4, Supplementary Information Text S4 and Supplementary Data 3). Moreover, G2a2* is represented by only one sub-lineage (G2a2b2a1a1b*, L497) except for the Iceman, which carries a different one (G2a2a1a2a1a1b, G-Z6208, or formerly G2a2a1a2 L-91)[36]. Comparative analyses (AADR v54.1 dataset)[50] show that the G2a-L497 has been found especially in EN (Linearbandkeramik, LBK) individuals from Germany and MN individuals from France[22,51], while the Iceman's lineage was more widespread. Indeed, it has been found in EN from Germany (LBK) and Spain, MN from France and Croatia, and also in more recent MBA individuals from Croatia[22,52]. CA individuals from North Italy (Broion) carried a different lineage of G2a* (G2a2b2b-F705 or former G2a3-F705)[42], while the Y-Chromosomal haplogroup in the other CA northern Italian males (Remedello) is not available for comparison (low coverage data)[7].

Among the alpine MN individuals, ISE01 (4445–4261 BC) is the only one with a different Y-Chromosomal haplogroup (J2a2a*). This haplogroup was not found in the comparative dataset and appears to be rare in ancient samples and mostly restricted to the eastern Mediterranean, Caucasus and Asia, where it may have originated by reaching western Europe during the NE[48,53,54]. ISE01 is the most recent among the MN analysed and belongs to an advanced phase of the VBQ culture (VBQ3) (Supplementary Information Text S1).

Instead, the Y-Chromosomal haplogroup R1b* of SIU01 is one of the principal haplogroup in the ancient Eurasian individuals but was found only in this alpine male (see Discussion).

Unlike alpine paternal lineages, maternal ones are composed of different macro mtDNA haplogroups (H, J, K, U, V, X), which are typically found in other prehistoric individuals from Europe (Fig. 4 and Supplementary Data 3). However, some alpine sub-lineages (e.g. H3k, H5d, J1c12b, V15, J1c + 1626 + 189) are very rare elsewhere or have not yet been observed according to the comparative dataset (AADR v54.1; samples with mtDNA "coverage >2× or published" and dated from -5500 to 1200 BC)[50]. Our comparative analysis also shows that the mtDNA haplogroup of ISE01 (T1a) was described in some NE samples from south-eastern and central Europe and in more recent samples from the Middle East (Israel, Armenia)[55–57] whereas that of SIU01 (X2b + 226) was found mainly in ancient individuals from central, northern and south-western Europe (MN to MBA)[41,58–60]. Finally, the K1f mtDNA haplogroup of the Iceman, never identified so far in modern and ancient individuals[61,62], was also not found in any of the other alpine individuals analysed in this study.

Kinship analyses clearly detected close relationships (parent-child or siblings) in at least four cases, and in other two cases, a 2nd degree relationship. Kinships always involved individuals from the same archaeological site and were buried in multiple or single graves. For instance, using information from unilinear transmitted markers, one possible mother-child (CA; MEZ01/MEZ02) and father-child (CA; ORA01/ORA01)[38] and two possible brother-sister pairs (NOG201/NOG202 from CA and VEL701/VEL705 from MN) were identified (Supplementary Information Text S4 and Fig. S17, Supplementary Data 5).

Furthermore, our ROH analysis did not reveal a high level of inbreeding resulting from union between first-degree cousins

(sROH$_{>20}$ over 50 cM) according to other studies[63,64]. Instead, it identified inbreeding stemming from a union between second-degree cousins in four individuals, including three males (COR01, NOG301, ROM308) and one adult female (SOL01). Additionally, ROH analysis also indicates a higher sum of runs of homozygosity (ROH > 4 cM = 87 cM) for our ME (MAD01) compared to other alpine sampled individuals, suggesting a smaller population size during ME in the EIAlp (Supplementary Information Text S5 and Fig. S18, Supplementary Data 9), as also described in other contexts[63,65]. Additionally, we found an increase in the effective population size (Ne) from MN (mean sROH$_{[4-8]}$ = 13.46 cM) to CA (mean sROH$_{[4-8]}$ = 4.38) cM followed by a slight decrease in Ne starting from the CA-EBA transition phase, with a similar trend for the following EBA and MBA periods (mean sROH$_{[4-8]}$ between 7.87 and 9.1 cM).

Regarding phenotypic traits analyses, full prediction (based on 41 SNPs) for hair, eye and skin colour was possible only in six individuals. These likely had brown eyes associated with dark brown to black hair colour (similarly to the Iceman) and skin from pale to intermediate skin colour (Supplementary Information Text S9, Supplementary Data 21). However, inconsistent results on Iceman's skin colour compared to the previous study by[37] based on high genome coverage, raise the doubt that the genome coverage in our study does not have sufficient power to correctly predict this complex trait (Supplementary Information Text S9).

In addition, some phenotypic traits that have been associated with a single or few SNPs and which have also been investigated in the Iceman were analysed in our study. For instance, the derived allele of the SNP *rs4988235* in the *MCM6* gene (*MCM6/rs4988235*) associated with lactose tolerance was absent in all prehistoric alpine individuals analysed. Moreover, the derived allele of the SNP *PLRP2/rs4751995*[66,67], possibly associated with an agricultural diet adaptation, was not detected in the alpine ME individual but was found from the MN onwards, with an increasing occurrence of 62.5% in the MN group and 75% in the CA group. In the later period, a decreasing occurrence was observed (CA-EBA; 50%), whereas the EBA groups only show the derived allele at this position. The analysis of a tag SNP (*rs1495741*)[68] of the gene *NAT2* whose "slow acetylator" variant has been hypothesised to be advantageous in agriculturalist populations[69–71] found the derived allele at this position in all alpine groups and in the Late ME individual (Supplementary Information Text S9, Supplementary Data 22).

## Discussion

We extended the genomic analyses to 47 new prehistoric individuals from the Iceman's territory covering a time span of ~5000 years (from ME to the MBA), providing a more detailed picture of the genomic structure of prehistoric individuals from this alpine area over time. This allowed us to investigate the extent of major demographic events in the EIAlp that occurred in Western Eurasia during Prehistory.

The tooth sample of the ME individual (MAD01) was recovered from a rock shelter in the Adige Valley, and our radiocarbon dating (6380-6107 BC) places it in the last phase of the ME, characterised by the Castelnovian industry (Supplementary Information Text S1). In general, little is known about the last phase of the ME in Europe and the transition from this period to the NE[65]. In fact, MAD01 is one of the rare Late ME individuals from Italy and southern Europe genomically analysed to date, in addition to a few other individuals from Sicily[44,47], making the data from our alpine ME individual particularly valuable. Moreover, the availability of genomic data from this local ME made it possible to use it in our genetic admixture models without resorting to samples outside of the territory. Genomically, MAD01 resembles typical WHGs (Villabruna) with a contribution from EHGs (16.4 ± 3.7%). The absence of Fournol-related ancestry in the alpine ME genome, which has been found in Upper Palaeolithic individuals from western and southwestern Europe[44], suggests that alpine ME has a genetic link

to the East and could be a representative of migrations from the South-East.

Regarding the EHGs-related ancestry found in MAD01, our finding agrees with a previous study, which has shown that ME individuals from Europe dated <7500 years ago often carried EHGs-related ancestry[44]. In addition, we found a rather early admixture time between WHGs and EHGs in the alpine ME (-13,700–8300 BC), one of the earliest in Europe according to Chintalapati et al.[72], adding a piece to the timing of these admixture events in southern Europe.

From an archaeological point of view, transalpine cultural contacts during the Late ME in the EIAlp[73] are well documented in northern Italy (Mondeval de Sora)[16], central Switzerland, the Dinaric Alps and the Balkans[74].

A substantial shift in ancestry related to migrations of EN farmers from Anatolia can be clearly detected in the genomes of our alpine individuals from MN (starting from at least -4600–4400 BC, VEL102). These genomes carry on average -87% of ancestry related to EN and -13% of local HGs-related ancestry. It is not feasible to verify the presence of these ancestries in early farmers, as no human remains from EN have ever been recovered in this alpine area so far. With our study, we provide the first genomic data on NE individuals from northern Italy.

In our study, we also discovered that the HGs-related ancestry proportion remains constant in the EIAlp from the MN onwards, similar to other contexts in Italy (e.g. Sicily and Sardinia)[39,45], but different from central Italy[40].

From an archaeological perspective, it is still unclear how the transition from a hunting and gathering economy to a productive one took place in the EIAlp. Analysis of the material culture suggests a gradual process of acculturation of the ME groups[17,24], with one example consisting of the finding of a small female deer antler figure (Venus of Gaban) with clear elements of both ME and NE traditions[75,76], while the presence of a gap so far observed in the Gaban site between ME and NE underlines the discontinuity[77,78]. However, radiocarbon dates from other archaeological contexts in the EIAlp suggest continuity in the presence of ME groups at least until 5200–5100 BC[79].

Overall, our study suggests (f4-statistics, qpAdm) a genetic contribution of local ME in alpine MN farmers, while there may have been some contribution from other HG groups before reaching the EIALp, but not from the Balkan area. Moreover, since according to the archaeological data the EN farmers reached EIAlp by -5100 BC, our estimated time of admixture between ME and NE groups (-from 6100 to 5100 BC) could also support both local and non-local admixture in this alpine region.

The alpine genomes from the following CA phase (-3350–2300 BC) resemble those from the previous period. We could confirm that the Iceman has high (>90%) and low proportions (<10%) of ancestry related to EN and HGs, respectively. However, we also revealed that this ancestral model is not peculiar to this individual but is found in all the other twelve (not related) CA alpine individuals analysed in this study. Comparative analyses found similarly low values of HGs-related ancestry in northern Italy (Broion and Remedello) during CA, supporting previous findings[37]. Notably, in our study, we additionally uncovered that the same ancestral model persisted in most of the individuals from the following CA-EBA and EBA periods, including one of the two most recent alpine individuals of our dataset from the MBA (STE05; 1663–1511 BC). These results are similar to those found in ancient Sardinia, where, albeit with a more extreme pattern, a genetic continuity has been observed from the MN onwards[45].

In the alpine territory, this finding is clearly visible at the Y-Chromosome level, since >90% of alpine prehistoric males of all chronologies (excluding the ME) carried the same G2a* paternal haplogroup, unlike those from other European regions. For instance, in southeastern Europe[80] Y-Chromosomal haplogroups varied substantially within the same chronological phase and among

chronologies. In this region, during the CA, the frequency of haplogroup G2a* was approximately 25% and several other lineages were present, while it has not been found in the most recent EBA individuals from the same areas. In central Europe (Czech Republic), the haplogroup G2a* was present in individuals from the NE together with other lineages and disappeared in individuals from EBA and BA, which carried most of the haplogroups R1b*, R1a* and I2*[81]. Moreover, in northern Italy, Y-Chromosomal lineage distribution varied more[42] and in the Po Valley area, an individual dated to 2194–1939 BC, roughly contemporary with the alpine EBA males with the G2a* lineage, bore the R1b* haplogroup[41].

G2a* was the dominant Y-Chromosomal haplogroup of the first NE farmers who migrated from Anatolia to Europe[82,83]. For instance, this haplogroup has been found in Early NE from Anatolia (Boncuklu, 8300-7800 BCE), in NE (Starčevo–Kőrös–Criş culture, 6000-4500 BCE) and MN individuals in Hungary and Croatia (Sopot culture, 5000 BCE) as well as in individuals associated with the LBK culture (5500–4500 BCE) in Germany[22,52,55,56,84]. From a paternal perspective, this haplogroup points to a genetic affinity between alpine MN and NE individuals from the Balkans and more northern areas.

Within the Haplogroup G* (with a frequency of <10% in Europe), the G2a-L497 lineage found in the EIAlp is the most frequent in present-day populations. It has been proposed that it originated in central Europe and reached Italy most likely by migratory flows from the north[85]. High frequencies of this lineage have been found in more isolated populations from the north-western and central Italy[86] and in present-day individuals from more isolated alpine valleys in North Tyrol (Austria, G2a-L497 above 40%), closer to the alpine area under investigation[87]. This previous study also suggests an old settlement of this lineage in the Tyrolean Alps and additionally proposes that this territory may have played a significant role as a source or at least as a transit route for G2a-L497 lineage in Europe[87]. These findings can be supported by our aDNA study, which attests to a long-term presence of this paternal lineage in the alpine territory starting at least from the MN (-4600–4400 BC). Intriguingly, the Iceman carried a different Y-Chromosome lineage of G2a* (G2a2a1a2a1a1b, G-Z6208*) that is very rare in present-day populations of Europe (<1%), including those from North Tyrol (0.5%)[87], apart from some areas such as Corsica and Sardinia[36].

Starting from the transition phase from CA to EBA, in the EIAlp we detected a signal of lower gene flow from groups with different ancestry than those associated with ENs and HGs. Indeed, the Steppe-related ancestry was detected in a young female (2402–2149 BC, LAS01), associated with the formative aspect of the Polada culture and in another two female individuals from the same period and cultural horizon, which retain on average -22% of the Steppe-related ancestry (Supplementary Information Text S1). Notably, based on currently available genetic data and radiocarbon dating from other ancient Italian individuals, we suggest that the Steppe-related component emerged in Italy in the EIAlp (at least from -2400 BC), later in northern Italy and Sicily (-2200 BC), and then in central Italy (-1600 BCE). Nevertheless, analyses of more prehistoric Italian samples are still necessary to reconstruct the timing of this complex event.

The highest percentage of the Steppe-related ancestry (36.1 ± 2.6%) was found in the only male in our dataset, showing this ancestral component in his genome (SIU01; 1601–1295 BC). This individual additionally carried the Y-Chromosomal haplogroup R1b* (R1b-P312), also described in northern Italy (Broion)[42] and that has been related with migrations of Bell Beaker males from central and western Europe[41] and brought a rare mtDNA lineage detected in the same European areas[58,59]. The cranium of SIU01 has been recovered in the more northern alpine area of the EIAlp (Isarco/Eisack Valley) and belonged to an individual whose cultural attribution is not yet fully defined[88] (Supplementary Information Text S1). Overall, our results on SIU01 suggest a possible different origin of this individual and a

genetic connection with transalpine prehistoric groups from central and western Europe. On the other hand, the rare occurrence of the Y-Chromosomal haplogroup R1b* in the EIAlp compared to other European areas indicates very low genetic impact from males carrying the Steppe-related ancestry in the alpine groups, even if more BA and MBA samples from this alpine territory are necessary to confirm this result.

During the same CA-EBA period, but later than the appearance of the Steppe-related ancestry and independent of it, another ancestry related to IN farmers (average of ~20%), reached the EIAlp, starting at least from ~2200–2000 BC (NOG302). It has been shown that IN-farmers related ancestry spread from the Mediterranean (Aegean) to western and northwestern Europe with an increase starting from the Iron Age[39,40]. Based on the distribution of the IN-farmers related component in the EIAlp as well as in northern, central, and southern Italy, we suggest that this component reached the alpine region from the Italian areas south of the Alps and/or from the Aegean region[39]. Finally, around the same time, another component related to CHGs, but not nested in the Steppe-related ancestry, is detected in only one female individual (ROM402; 2272–2041 BC), which may have ancestors from the Eastern Mediterranean[89].

In summary, our study, while highlighting occasional gene flow from groups of different ancestry and origin, mainly reveals limited genetic exchanges in the EIAlp from the MN to the MBA. This could indicate a relative isolation of the prehistoric alpine groups, similar to other more isolated areas in Italy (e.g. Sardinia), compared to those from other European regions (e.g. south-eastern or central Europe).

This picture seems to be broadly consistent with the overall cultural development pattern in the EIAlp. In fact, over the long period considered in this study, in this alpine area, there appears to have been a cultural evolution that evolved without interruptions or sudden changes, despite evident cultural contacts with even distant geographical areas. Indeed, the most noticeable characteristics of cultural changes seem to be more linked to the acquisition of new knowledge and technological skills (such as exploiting sources of copper procurement and making metal objects). Moreover, from an archaeological perspective, there is currently no clear evidence of cultural contacts with groups that have been related to the diffusion of the Steppe-related ancestry. It could be hypothesised that Bell Beaker groups may have played some role in the spread of this genomic component, but the cultural Bell Beaker phenomenon appears to be extremely rare in EIAlp[90]. The diffusion of this component could possibly also be linked to the spread of the statue steles, diffused throughout Europe and also found in the EIAlp[91,92]. However, so far, clear funerary attestations related to such cultural manifestations are mostly lacking in this territory.

This study additionally found marked differences between the two uniparentally transmitted markers, with the Y-Chromosome showing very low diversity compared to mtDNA. Such differences have been observed globally as the result of a post-Neolithic Y-Chromosome bottleneck that may have occurred in Europe around 3000–5000 BP[93]. This pattern could be explained by the practice of some cultural rules in past communities (e.g. patrilinearity and patrilocality). In the EIAlp, the observed genetic pattern could be explained by patrilocality, which is when females moved to the male's birthplace where they had offspring, as has been observed in other prehistoric contexts in Europe[59,94]. In fact, the Y-Chromosomal lineage in the EIAlp suggests long-term and stable settlement for males. However, it cannot be excluded based on the available data that the males analysed came from another settlement in this alpine region. Additionally, most of the males resulted as not being closely genetically related and do not show strong signals of inbreeding, indicating a complex scenario in the EIAlp.

Regarding the phenotypic analysis, it suggests that, like the Iceman, all prehistoric alpine individuals were unable to digest milk after early childhood, a prevalent condition in ancient individuals and in modern populations, including farmers and pastoralist groups[95]. Moreover, based on the analyses of SNP *rs1495741* in the gene *NAT2*, we suggest that all the alpine prehistoric individuals analysed probably had high concentrations of fatty acids of plant origin, associated with a predominantly plant-based diet, as also observed in the Iceman[37]. Although the presence of this variant in our ME individual might seem unexpected, our results agree with what has been observed in a previous study that showed no changes in the allele frequency of the derived allele of SNP *rs1495741* over the past 10,000 years (HGs, EN and group with Steppe-related ancestry)[67]. Moreover, in general, the results in the alpine groups for the SNP *rs471995* of the *PRLP2* gene might suggest, in the EIAlp, from MN onwards, an adaptation to a diet based on cereals typical of farming communities. In fact, the variant of this SNP has been found to be relatively common in present-day populations with a cereal-based diet and more frequent in EN farmers than HGs, although no evidence of selection has been found[66].

To further investigate social practices with possible extended kinships, phenotypic traits and additional analyses such as metagenomics in past alpine groups, deep shotgun sequencing of the sampled individuals can be performed with the available biological material.

Finally, our study also uncovers that, although the Iceman presents the same ancestral genomic model and a similarity in phenotypic traits to the other CA sampled individuals (and with most of the other alpine prehistoric individuals analysed), he differs from them in his maternal and paternal lineages. These results suggest a slightly different genetic history for the Iceman compared to the other CA alpine individuals. Although stable isotope analyses of Iceman's teeth and bones allowed his origin to be pinpointed to a few valleys close to the discovery site (Schnals/Senales valleys)[96] we have no information on the cultural group to which the Iceman may have belonged, as he was recovered without any distinctive material elements (e.g. pottery). Our findings leave open some questions about the genetic origin and cultural affiliation of this enigmatic individual.

## Methods

### Authorisations by local authorities

Authorisations with prescriptions were obtained for the study, sampling, and temporary transfer of the human osteological finds owned by the Autonomous Province of Trento (Prot. n. 156 15.03.2021) and the Autonomous Province of Bolzano/Bozen (Prot. Nr. 636137-25.09.2019), as well as for genetic analyses, in the context of the project 'Genomic diversity of prehistoric individuals from the Iceman's territory in the Eastern Italian Alps' (see also Supplementary Information Text S1, Data availability section and the Reporting Summary). The human remains were collected from various Institutions scattered throughout the territory (after further authorisations by the specific Institution), which store them in their repositories. These include: MUSE of Trento, Castello del Buonconsiglio of Trento, Museo di Scienze e Archeologia- Fondazione Museo Civico of Rovereto, Ufficio beni archeologici of the Soprintendenza per i Beni Archeologici della Provincia Autonoma di Trento and the Ufficio beni archeologici of the Soprintendenza Archeologica della Provincia Autonoma di Bolzano/Bozen.

### Molecular analyses and radiocarbon dating ($^{14}$C)

Pars petrosa (PP; $N = 43$), which were isolated or still articulated with the skull, and tooth samples ($N = 9$) were selected for genetic investigation in the laboratories of the Institute for Mummy Studies of Eurac Research in Bolzano (Italy) (Supplementary Data 2). After detailed photographic documentation (before and after sampling), approximately 200 mg of powdered bone and dental root were collected. For the PP, a thin-tipped drill was used to create a small hole in the cortical bone that forms the structure of the cochlea (part of the osseous labyrinth of the inner ear)[97]. This approach minimised the invasiveness

of the sampling, avoiding the cutting and the destruction of the bone and making it accessible for further (e.g. macroscopic) studies. Regarding the tooth sampling, the root was separated from the crown using a Dremel and then milled. Using the root, the pulp chamber was also included, which preserves microbial DNA, making DNA samples available for further investigations.

The sampling was conducted in a dedicated pre-PCR area of the aDNA laboratory following all stringent rules required for aDNA analyses. DNA samples were then extracted using a silica-based method[98] and double-stranded genomic libraries were constructed[99,100]. These were then sent to an external company (Macrogen) for shotgun sequencing (HiSeq-X system, Illumina). The specifications of the reagents used, company names and catalogue numbers can be found in Supplementary Data 23.

The samples (with more than 1% of HR) were then enriched for ~1.24 million SNPs across the human genome[48] as well as 49 K additional sites on the Y-Chromosome and the complete mitogenome using the in-solution target capture kit myBaits ® Expert Human Affinities−Prime Plus (Arbour Bioscience) (Supplementary Information Text S8).

Radiocarbon dating ($^{14}$C) was carried out on 36 samples, and only two did not have sufficient preserved collagen for the analysis (Supplementary Data 1). The remaining samples were calibrated (sigma 2) with a 95% probability, except for one (ROM308), whose calibration accuracy did not exceed 68% (Supplementary Information Test S2 and Fig. S16).

The raw genetic data generated in this study (shotgun and capture) are available at the European Nucleotide Archive (ENA: https://www.ebi.ac.uk/ena/browser/home) with the accession number PRJEB70242.

## Sequence read processing, alignment and damage patterns

The raw sequence data were processed by first trimming and merging reads using PEAR (v.0.9.10)[101] with a minimum length of the assembled sequences and a minimum number of overlapping bases of 25 bp. Merged reads were then mapped to the Genome Reference Consortium Human Build 37 (hg19) and to the mtDNA reference genome (rCRS)[102] using BWA (v.0.7.17)[103]. We then converted data into BAM files using SAMtools (v.1.16.1)[104] with minimum mapping quality set to 30. Duplicates were removed with Dedup[105], and mapDamage2 (v.2.2.1)[106] was used to authenticate ancient DNA by assessing the damage patterns in all our samples. To evaluate the number of SNPs sites hitting the 1240 K panel[48], vcf files were created by using BCFtools (v.1.16)[107].

## Contamination estimates

Contamination levels based on mtDNA data were estimated in all individuals using Schmutzi (v.1.5.6)[108]. For the male individuals only, estimates were based on X-Chromosome data using the method implemented in ANGSD (Analysis of Next Generation Sequencing Data; v.0.941)[109]. When the contamination level was above 5%, the PMDTools method (threshold set to 3) was applied to filter the sequences identified as genuinely ancient (v.0.60)[110].

## Genotyping

All bam files were genotyped by using pileupCaller (https://github.com/stschiff/sequenceTools) and pseudohaploidized by randomly calling an allele for each position of the 1240 K panel (1,233,013 SNPs)[48]. We removed indels and the transition sites to avoid taking possible false polymorphisms due to deamination into account in the analyses.

## Molecular sex determination

The molecular sex of all individuals was assigned using the method described in ref. 111, which is based on the number of reads mapping to the non-recombining portion of the Y-Chromosome and X-Chromosome compared to the autosomal reads, and the method described in ref. 112 that analyzes only X-Chromosome linked markers.

## Mitochondrial DNA and Y-Chromosome haplogroups assignment

Consensus mtDNA sequences were obtained for each individual using Schmutzi (v.1.5.6)[108] with a minimum mapping quality of 30. The obtained fasta files were used for the mtDNA haplogroups assignment using HaploGrep 2 (v.2.4.0)[113]. Instead, Y-Chromosomal haplogroups were assigned to male individuals using Yleaf statistical software version 2.2[114] on the bam files, with minimal base-quality of 30 (-q 30) and base-majority to determine an allele set to 90% (-b 90).

## Run of homozygosity (ROH)

ROH values were inferred by using the software hapROH (v0.63)[63]. The default parameters and pseudo-haploid data as input were used. ROH was determined using the default genetic map of the software and 5008 haplotypes from the 1000 Genome as a reference panel[115] and SNPs were called for each chromosome and individual (Supplementary Information Text S5).

## Kinship analyses

Three different methods were applied (READ, TKGWV2 and KIN) to infer possible genetic relatedness between the alpine individuals analysed (Supplementary Information Text S4). For READ (Relationship Estimation from Ancient DNA (v.1)[116], we use pseudohaploid genomes with only transversions for all our samples, and the normalisation has been done based on the median of all average P0s. For TKGWV2 method (Two-Sample Kinship and Genetic Relatedness with Haploid and Diploid Genomes (v. 2) we followed the pipeline of Fernandes and collabotarors[117] (https://github.com/danimfernades/tkgwv2) to generate transposed PLINK text files for all pair of individuals and we used the allele frequencies file provided by the author, based on modern European CEU data from the 1000 Genomes project Phase 3. Lastly, for KIN (v.3.1.3), the software KINgaroo was used to generate the input files from the bam files previously generated[118]. For contaminated samples (>5%) filtered by using PMDTools, we did not utilise the contamination correction (-cnt:0) nor the estimation of the location of long runs of homozygosity.

## Datasets

A dataset of genomic data from 996 present-day individuals from the Human Origins (HO) panel[1,119] was compiled and merged with data from our alpine individuals. In addition, 1.341 West Eurasian ancient individuals were merged with published data from the Allen Ancient DNA Resource (AADR v54.1)[50] and used for the PCA and ADMIXTURE analysis (Supplementary Data 10). We filtered ancient individuals in the dataset by removing those with less than 20,000 SNPs and a coverage <0.01. Among duplicated and related, only the individual with the highest number of SNPs was kept. Finally, individuals considered as QUESTIONABLE were not considered. The overlap between the 1240 K SNPs panel[48] and the HO data[122] was 593,054 SNPs. After removing indels and transitions, we obtained a total number of 110,416 autosomal SNPs. A second dataset was obtained for qpWave, qpAdm and $f$-statistics analyses by merging our individuals with the 1240 K AADR dataset (v54.1) (1240 K SNPs)[50] using the same set of individuals selected for building the dataset used for PCA and clustering (ADMIXTURE) analyses (without HO data, Supplementary Data 10), with additional 114 recently published ancient genomes[44] (Supplementary Data 11).

## Principal components analysis

Principal components analysis (PCA) was performed using the programme smartpca (v16000) (EIGENSOFT)[120,121] with parameters shrinkmode: yes; lsqproject: yes, on the HO dataset. We performed

analysis on modern West Eurasian individuals on which ancient genomes were projected. The period of all the individuals in the PCA was redefined based on the time periods for EIAlp (ME > ~5500 BC; NE: ~5500–3500 BC; CA: ~3500–2200 BC; CA/EBA: ~2900–1900 BC; EBA: ~2200–1500 BC; MBA: ~1600–1300 BC).

## Clustering analysis

The software ADMIXTURE (v.1.3.0)[122] was run after pruning, to remove SNPs that are in strong linkage disequilibrium, using the option −indep-pairwise 200 25 0.2 in PLINK (v.1.9)[123]. The software was run from $K = 2$ to $K = 12$, with 10 replicates with random seed values for each $K$. We used the option −cv (cross validation) to find the $K$ value with the lowest errors (Supplementary Information Fig. S21). Results for $K = 5$ with the lowest cross-validation value were plotted using Pong (v.1.4.9)[124] (Supplementary Information Fig. S22).

## qpWave/qpAdm analyses

Pairwise-qpWave/qpAdm was run from ADMIXTOOLS (v.7.0.2)[8,119] on the 1240 K SNPs dataset. The analysis was done with default parameters and allsnps: YES. We first run pairwise-qpWave to test pairwise clustering of our prehistoric alpine individuals and determine if those from the same period could be grouped ($p$-value > 0.01). We used a set of right population (outgroup) including, Mbuti.DG, Onge.SG, Papuan.DG, Han.DG, Karitiana.DG, Ethiopia_4500BP.SG, Russia_Ust_Ishim.DG, CHG, Italy_Villabruna, Spain_ElMiron, EHG, Iran_GanjDareh_N, Russia_MA1_HG.SG, Russia_Kostenki14.SG, Israel_Natufian, Czech_Vestonice, GoyetQ116-1, Hungary_Koros, Levant_N, Anatolia_N, Russia_Samara_EBA_Yamnaya. A heatmap based on the $p$-values calculated with pairwise-qpWave for each pairwise was generated using heatmap.2 function of the R-package gplots[125] (Fig. 3 and Supplementary Information Fig. S23).

We then run qpAdm to estimate the ancestral model of prehistoric alpine individuals with the R package admixr (v.1.0.0)[126]. We used as right populations (outgroups): Mbuti.DG, Onge.SG, Papuan.DG, Han.DG, Karitiana.DG, Ethiopia_4500BP.SG, Russia_Ust_Ishim.DG, CHG, Italy_Villabruna, Spain_ElMiron, EHG, Iran_GanjDareh_N, Russia_MA1_HG.SG, Russia_Kostenki14.SG, Israel_Natufian, Czech_Vestonice, GoyetQ116-1, Hungary_Koros, Levant_N, Anatolia_N (Supplementary Data 16-19). When one of these populations was used as a source, it was removed from the right population.

For our ME individual (MAD01), we performed two-way admixture models using Italy_Villabruna, EHG, OKL, GoyetQ2 and Fournol as sources (Supplementary Data 16).

For all the other individuals from MN to MBA, we performed two-way and three-way models using different combination of sources: MAD01, Italy_Villabruna, Anatolia_N, EHG, Iran_GanjDareh_N (Iran_N) and Russia_Samara_EBA_Yamnaya (Steppe), or a four-way model (ROM402, Fig. 4) using MAD01, CHG (Caucasus HGs), Anatolia_N and Russia_Samara_EBA_Yamnaya (Steppe) (Supplementary Data 17 and 18).

We also performed the qpAdm analysis on groups defined based on the pairwise-qpWave/qpAdm analysis and chronologies (Supplementary Data 19). qpAdm analysis for 2-way models in the alpine groups or individuals (PrehAlps_MN, PrehAlps_CA, PrehAlps_CA-EBA, PrehAlps_EBA PrehAlps_STE05) was also done by using a group of WHGs (Hungary_Koros; Germany_Blatterhohle; Switzerland_Bichon.SG; Croatia_VelaSpila; Italy_OrienteC; Falkenstein; BerryAuBac; Iboussieres25-1; Iboussieres31-2; Rochedane; Drigge, MAD01), instead of MAD01 alone as a source of HG-related ancestry (Supplementary Data 19). We considered the most parsimonious model with a $p$-value > 0.01 to be most likely.

## f-statistics

All $f$-statistics were performed with the software ADMIXTOOLS (v.7.0.2)[8,119] and the package admixr (v.1.0.0)[126] in R with default parameters.

For the $f_4$-statitsics, we used the parameter allsnps: YES. We computed a $f_4$-statistics of the form $f_4$(pop1, pop2; test, Mbuti) to trace possible genetic affinity between our MN individuals (test) and others NE individuals from Europe (pop1 and pop2) or between alpine MN (test) and different groups of HGs and MAD01 (pop1 and pop2) (Supplementary Data 15).

Outgroup $f_3$-statistics analysis of the form of $f_3$(Mbuti; pop1, pop2) was also performed (Supplementary Data 12). To visualise the genetic affinity of MAD01 with others HGs individuals we executed a MDS analysis based on the dissimilarity matrix of pairwise genetic difference values $(1 - f_3)$ and we then used the function cmdscale from the package stats (v.3.6.2, https://www.R-project.org) (Supplementary Information Fig. 2B). The genetic clusters were defined in the MDS based on the Extended Data Table 1 and the Fig1c from[44]. In addition, $f_3$-statistics of the form: $f_3$ (Mbuti; European pop, Alpine group) were performed to estimate possible genetic affinity between European populations and alpine groups, from MN to MBA, as defined by qpWave/qpAdm analysis (Supplementary Data 13 and Supplementary Information Fig. S24).

## Estimates of admixture time (DATES)

We used the linkage-disequilibrium-based method DATES (v.4010)[53,72] to estimate the timing of admixture between ancestral populations MAD01, Anatolia_N and Steppe in the alpine groups or individuals (defined on qpAdm analysis). We consider 28 years for one generation. To estimate the admixture time between WHG and EHG groups in the alpine ME (MAD01), Villabruna or a group of WHGs (Luxembourg_Loschbourg.DG; Iraly_Villabruna; Spain_Labrana; Switzerland_Bichon.SG; Italy_OrienteC) was used as a source for WHGs (Supplementary Data 20), and the model was verified by qpAdm analysis (Supplementary Data 16). Instead, to estimate the admixture time between HGs and Anatolian farmers (Turkey_N) in alpine groups or individuals, we used MAD01 or a group of WHGs (Hungary_Koros; Germany_Blatterhohle; Switzerland_Bichon.SG; Croatia_VelaSpila; Italy_OrienteC; Falkenstein; BerryAuBac; Iboussieres25-1; Iboussieres31-2; Rochedane; Drigge, MAD01) as a source for WHGs, and the model was validated by qpAdm (Supplementary Data 19 and 20).

## Phenotypic analysis

Hirisplex panel (https://hirisplex.erasmusmc.nl/) was used to attempt to predict eye, skin and hair colour in the alpine individuals. We also analyse specific variants that have been analysed in the Iceman[37] and are associated with phenotypic traits of interest (Supplementary Data 21 and 22). Target regions were extracted with bcftool mpileup, and variant-calling was done with bcftools call (v.1.16)[107]. We filtered variants for a base quality of 30 and with a minimum coverage of 5 reads. The allelic frequency was calculated with PLINK (v1.90)[123].

## Reporting summary

Further information on research design is available in the Nature Portfolio Reporting Summary linked to this article.

## Data availability

The raw data (FASTQ files) generated in this study have been submitted to the European Nucleotide Archive (ENA: https://www.ebi.ac.uk/ena/browser/home) with the accession number PRJEB70242. Access codes to previously published data: AADR v54.1 database [https://reich.hms.harvard.edu/datasets] and dataset from [https://doi.org/10.1038/s41586-023-05726-0; ENA accession number PRJEB51862]. The osteological remains of the individuals sampled in this study are under the protection of the Autonomous Province of Bolzano/Bozen and the Autonomous Province of Trento and are preserved at the following local authorities (contact person and Director at the time of the study are listed in brackets): the MUSE of Trento (Alex Fontana and Michele Lanzinger), Fondazione Museo Civico di

Rovereto (Maurizio Battisti and Alessandra Cattoi), Museo del Buonconsiglio di Trento (Morena Dallemule and Laura Dal Prà), Ufficio Archaeologico, Provincia Autonoma di Bolzano/Bozen, Italy (Catrin Marzoli), UMSt Soprintendenza per i beni e le attività culturali, Ufficio beni archeologici, Provincia Autonoma di Trento (Elisabetta Mottes and Franco Nicolis,) and the Department of Humanities, University of Trento (Annaluisa Pedrotti, Marco Gozz).

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

## Acknowledgements

This research received funding from the "Research Südtirol/Alto Adige 2019" from the Autonomous Province of Bolzano/Bozen (Grant Agreement no. 16170/2020). Additional support was provided by the European Regional Development Fund 2014-2020_CALL-FESR 2017 Research and Innovation_Autonomous Province of Bolzano-South Tyrol (Project: FESR1078-MummyLabs). One-year postdoctoral scholarship to M.C. was funded by the Sven and Lilly Lawski Foundation. The authors thank the

Fondazione Museo Civico di Rovereto, Museo del Buonconsiglio di Trento, the MUSE of Trento and, the Ufficio Archeologico, Autonomous Province of Bolzano/Bozen, Italy (Catrin Marzoli). We thank Katrin Renner (Eurac Research) for editing the map in Fig. 1 and Viviana Conti for editing Fig. S16 and for the English revision of the Supplementary Information. The computational results of this work have been achieved using the Life Science Compute Cluster (LiSC) of the University of Vienna. We are grateful to the Department of Innovation (Research University and Museums of the Autonomous Province of Bolzano/Bozen) for covering the Open Access publication costs.

## Author contributions

Conceived the study: V.C. with contribution from A.Pa. and A.Pe.; Analysed the data: M.C. with input and supervision from V.C. and T.G; Organised the sampling collection: A.Pa. with V.C. and A.Pe., Selected and sampled the human remains: A.Pa.; Provided archaeological context: A.Pe., A.Pa.; E.M. and F.N.; Performed the molecular analyses: S.A. and S.Z.; Wrote the original draft: M.C., A.Pa., A.Pe., V.C.; Final draft: V.C. with inputs from all coauthors; Supervision, Funding acquisition and Project Administration: V.C., Additional found resources: F.M.; A.Z.

## Competing interests

The authors declare no competing interests.
