## [Transparent Peer Review file · Nature Communications]

Genomic diversity and structure of prehistoric alpine individuals from the Tyrolean Iceman's territory

Corresponding Author: Dr Valentina Coia

Version 0:

Reviewer comments:

Reviewer #1

(Remarks to the Author)

Manuscript #NCOMMS-24-36149, Croze et al., Genomic diversity and structure of prehistoric individuals from the Eastern Italian Alps: Insights from the Tyrolean Iceman's territory.

This manuscript from Croze and collaborators entitled "Genomic diversity and structure of prehistoric individuals from the Eastern Italian Alps: Insights from the Tyrolean Iceman's territory" provides wide-genome data for 47 new individuals from South Tyrol, and 34 new radiocarbon dates, from the Late Mesolithic to the Early Bronze Age. The Mesolithic individual is the only Castelnovian individual known to date on the European continent, and the individual LOS01 is the earliest individual with the Steppe-ancestry component in Italy, bringing great new data to the big picture of European dynamics.

I fully support the publication of this manuscript, in which the analyses are sound and robust, and the interpretations and conclusions well supported.

However, I have some comments, major and minor, mainly to help improving the manuscript, see below.

Major comments

- The introduction needs a bit of work for a clearer description of the context in a European perspective (Neolithic diffusion for instance). Some concepts are missing which are used later in the discussion.

- Page 6 paragraph 2 and P11 para3. Is this because of the absence of the Fournol ancestry in MAD01 that is also absent from Villabruna but not from Oberkassel? Have you tested it? Because if so, you would have a great hit that your Late Mesolithic Castelnovian individual has connection with the East and could be a representant of recent migrations coming from the South-East, that we haven't been able to show so far by lack of individuals in southern Europe. That would be a very interesting lead to explore in your paper. Given the importance of this Mesolithic sample as the only representant of the Castelnovian culture in southern Europe (except for the Uzzo individuals but Sicily has a different and further history), I think this deserves more digging.

- P7 para1 and P12. You observe a similar amount of HG ancestry through time in your samples from the Neolithic on, with no "resurgence" as observed in central Europe for instance. You say that MAD01 is the best fit, based on qpAdm 2-way models with alternatively MAD01, Drigge and Villabruna, for the pre-Steppe individuals. I am wondering several things here. Given the constant amount of HG ancestry, another plausible scenario would be that the HG ancestry has been carried on by the first migrant farmers from further East and stays stable through time, with no local admixture. I don't think the qpAdm models can rule out this hypothesis, as the p-value doesn't reflect a "better" fit if higher. I think you should try to disentangle this question using f4-tests and see if you obtain different results according to different HG proxies.

Also, the admixture date estimate raises questions here. I find it strange that you got no or poor success applying DATES, and I would consider running it with a group of HG, instead of only one individual (cf. Chintalapati et al 2022

<https://doi.org/10.7554/eLife.77625>), this improves the inference. Precise results on the admixture date would help the discussion about local vs. non-local HG component. The sentence written P12 para 2 "Analysis of the material culture suggests a gradual process of acculturation of the ME groups^{13,10} while the presence of a gap in the dating between ME and NE underlines the discontinuity^{72,73}." would also go in the sense of no contacts locally, according to the radiocarbon dates.

If different proxies are used and work for DATES, I would suggest to rerun the qpAdm accordingly as well, to have a

consistent set of analysis.

- Also, why didn't you apply an IBD sharing approach to your analysis, for instance with anclBD (Ringbauer et al 2023, <https://www.nature.com/articles/s41588-023-01582-w>)? You have such a good coverage overall that it would be very interesting to explore the relationships between your groups and the neighboring groups through time, to explore more in depth the geographic networks. This would also be very interesting for the Neolithic individuals, to disentangle the links with the North vs. the South of the Alps, even if the number of samples and their coverage in the South would be limiting.

Minor comments

Abstract

52 individuals are mentioned in the abstract, but only 47 are used in the study. I would make it clear from the abstract, to avoid any kind of "misleading advertising".

I would precise which HG groups you talk about for the admixture of the Mesolithic genome.

The differences of the Iceman being exclusively linked to uniparental haplogroups, I would make this clear here. Otherwise, it is unclear that you use his genome to compare the others, speak about "continuity" and then state he is different...

Introduction

- Overall, double check the calls to the references, some commas are missing, and the numbers are unclear (e.g. 34 instead of 3,4). Also, double check the reference list, some of them are wrongly formatted (e.g. 45, 77, 88...).

- P3 para1. You mention the Trentino-Alto Adige region from the first sentence, but don't state why it's important for the reader to know that. Specify that's the target of your study.

- P3 para2. You don't write "Neolithic" in the first sentence. You have to properly define it before using the term afterwards.

- P3 para2. Name the "Impressa" group with its name. Overall, we miss the big picture here: for the Neolithic, we need to know that two currents are present at the European level, to fit in with the local data.

- P3 para2. "From the mid-4th mill. BC, within the CA (3500-2200 BC), significant social changes occurred in the EIAIp with the formation of cultural groups distinguished mainly by different funerary practices": which ones? What is different?

Results/Discussion

- P5 para3. Define "1240K" the first time you mention "1.24 million SNPs".

- P5 para3. When you mention the related individuals, mention how many of them (or how many pairs).

- P6 para1. In the first sentence, maybe add "younger" when you talk about "all other alpine samples", or "as expected for a Mesolithic individual": it's so obvious that it took me a while to understand why you were saying this...

- P6 para4. You never explain why you don't have local EN (or earlier Neolithic) samples. Are the remains missing in the region? Because in the introduction you state that the Neo starts around 5100BC, but your earlier individual is from 4600BC.

- P8 para2. In the first sentence, why "may explain"? What other explanation would it be for the position of this individual on the PCA? I think you can use "does" or directly "explains".

- P7-8. In the whole section about CA-EBA, I would mention here that you have tested the relatedness between your samples and the published ones and didn't find any close one. You explain it in the SI, but I would add this here.

- P12-13. I'm surprised not to find any discussion about the YC haplogroup G2a, as a marker of affinity between the Alpine individuals and the Balkans or more Northern regions for the Neolithic.

- P13 para3. Why is this paragraph here? It is about Neolithic, I don't understand why it is not with the discussion about Neolithic P12, it's confusing to me and make the discussion harder to follow.

- P15 para3. "Moreover, from an archeological perspective, the appearance of the Steppe-related ancestry in the EIAIp is rather unexpected, as there is currently no clear evidence of cultural contacts with groups from the Steppe region." This is extremely simplistic... I don't believe your intention is to mean that actual Steppe groups came into direct contact with the EIAIp, and I think this needs to be rephrased.

- P15-16. At the end of the page 15, beginning of 16, given the number of available individuals, I would tone down a bit the interpretation of kinship practices and population structure.

- P16. How do you explain the derived allele at position rs1495741 in the gene NAT2 for the Mesolithic individual? There is no discussion about that, though it seems to be an unexpected result. Please develop.

Material/Methods

- P17. In the section "Molecular analyses and Radiocarbon dating (14C)", you mention that the samples were enriched for "more than 2 million polymorphic sites in the human genome using the in-solution target capture kit myBaits® Expert Human Affinities – Prime Plus (Arbor Bioscience)". What are these 2 million SNPs? You didn't use the kit "ancestral" (or I don't remember the exact name, as it is not commercialized anymore) which would have had more SNPs, but keep using the 1240k SNPset in your analysis. I'm confused here.

- P17. In the section "Molecular analyses and Radiocarbon dating (14C)", you mention 37 minus 2 radiocarbon dated individuals, so 35, when you say 34 in the introduction. Please doublecheck your numbers.

- P17. About the contamination + quality control, I would tell the number of individuals excluded from the analysis in the main text.

- P19. In the section "Datasets", the minimum threshold used for the analysis is 5,000 SNPs, that's very low. I don't know if there is a consensus about that, but usually the limit is set at 20,000 SNPs on the 1240K dataset... Can you justify why you went that low? Didn't it affect the PCA? The analysis? According to my experience, I think it does...

- P20. In the section about qpAdm, you don't mention the four-way model used in the main text and in Figure 4 for ROM402. Also, why do you set the p-value threshold at 0.01, and not 0.05 like commonly used? Then of course, most of your models in qpAdm work...

- P21. For DATES, see my comment in the results section.

- Tables S2 and S3: I found the two columns “# 1240K Sites (All)” and “# 1240K SNPs (All)” very confusing by their labels. Why do you need the second one, when this label is usually used for what you mean here in the first one? I would suggest simplifying, or making it clearer what you mean in the column headers maybe?

- Text S6. The discussion here about the two currents of Neolithic diffusion would have a better place in the main text.

- Figure S29. There is an error in the FALSE/TRUE colors in the 4th plot on the first line.

Reviewer #2

(Remarks to the Author)

Croze et al generated and analyzed new ancient DNA data from 47 individuals from the Eastern Italian Alps (also referred to as “the Tyrolean Iceman’s territory”), dating from the Mesolithic to the Middle Bronze Age. The main objective is to investigate how the genomic structure varied over time in this region, and incidentally bring more context to the famous Iceman. They also report 34 new C14 dates.

Croze et al first screened the NGS data obtained from ancient DNA extracts to assess DNA preservation in the vestiges, then performed in-solution target enrichment for the libraries with >1% human DNA content, followed by DNA sequencing. They investigated migrations events, admixture between local hunter-gatherers and incoming farmers, timing of the arrival of people with Steepe-related ancestry, biological relatedness and a few phenotypic traits. The main conclusion is a relative stability and isolation of the prehistoric alpine.

The analysis performed are all in line with what is used “in routine” for human ancient DNA studies, but applied here to a new original dataset. No new type of analysis were performed, including for phenotyping for instance, where the few phenotypic traits examined have already been explored in previous studies.

It somehow reads like the authors do not have a strong research question, apart from filling a gap in the ancient human datasets (indeed, few ancient human DNA have been sequenced in this region). They do not developed any new method, nor proposed any improvement in previous methods.

My main concern is about the methodology followed to generate the data. The authors have access to a large and interesting set of human vestiges. Following screening, it appeared that many extracts are very rich in human DNA (up to 75%!), including the rare individual from the mesolithic (20%). The choice of performing capture in such a case, and not doing shotgun sequencing of whole genomes is extremely questionable. This is even more questionable when the authors are aware of the biases introduced by such capture, and aware of how exceptional the remains they had access to are (page 11: “little is known about the last phase of ME in Europe...MAD01 is one of the rare late ME samples from Italy and Southern Europe genomically analyzed to date... making the data from our alpine ME sample particularly valuable”, page 12: “These are the first genomic data available on NE samples from Italy”, page 13 “ISE01 represents the only burial found in northern Italy attributable to this cultural context”). The authors themselves maintain some confusion between “genome” (term usually used for whole genome sequencing at at least 1x), and what they call “genomic data” (which is actually a set of captured SNPs). By performing capture, the authors lost a vast amount of very valuable genomic information (especially for the rare mesolithic individual), and also metagenomic data (especially for DNA extracted from teeth), that could have been extensively re-used in future studies.

It is not clear how the libraries were built. The reference for library protocol dates from 2010. Is it exactly this one that was used? If I am correct, this protocol is single-indexed. Then how did the authors ensure their data did not suffer from index-hopping.

The vast majority of the conclusions are presented in a cautious way and are supported by the data. Some would need to be better justified, or toned down. For instance:

- Page 15: “the origin of SIU01 outside the territory of EIALp” Only isotopic data could help answering this

- Page 15: “we suggest that the observed genetic pattern could be explained by patrilocality”. The data cannot support this hypothesis. All males generated have the same Y haplogroup, therefore it is not possible to observe, with this type of data, if males were incoming from another settlement from this region. In addition, as the authors state themselves page 16 “most of analysed males resulted as not being closely related”

The sentence page 15 “the union between second degree cousins... was found to involve especially males (3 out of 4)” does not make sense to me. Said males are the offspring, they are not involved in the union.

About the archaeological description in the supplementary data: It is sometimes difficult for the reader to pull out the information that is useful for understanding the paper conclusions. In this section, we suggest the authors specify the location where the vestiges are currently stored, and which authorization (from which legal authority, permit number...) they received to perform destructive analysis.

It is unclear how the individuals have been classified as EBA; EBA/MBA or MBA. Is this classification based on archaeology or C14 dating? In particular, STE01 is EBA, STE03 is EBA/MBA and STE05 is MBA, while the 3 have overlapping C14 dates, STE03 and STE05 are from the same burial, and the 3 of them are related. This should be clarified, especially as the supplement states it is MBA, with a different date.

About the determination of the individual age: in the supplementary tables, when the authors state “revised for the present study”, please specify which method was used to determine the age at death.

As a general note, we should try to refrain from using the term “sample” when writing about vestiges from deceased individuals, as some people may find it offensive and disrespectful.

Version 1:

Reviewer comments:

Reviewer #1

(Remarks to the Author)

Manuscript #NCOMMS-24-36149A, Croze et al., Genomic diversity and structure of prehistoric individuals from the Eastern Italian Alps: Insights from the Tyrolean Iceman's territory.

I want to thank the authors for their efforts answering my comments, and for the extra analysis provided.

I have a few additional comments before fully validating the manuscript for publication.

- Typos are still present, such as “LBK” at the end of a paragraph for no reason, page 3. Please double check your manuscript.

- Page 7: I appreciate the effort concerning the additional f4-tests, but I think a rephrasing is needed here. You cannot talk about “negative” results if you don’t provide the structure of the f4 test. Here you need to write “... an f4 statistical analysis on the form f4(HG_test, MAD01; MN, Mbuti) was performed with different sources of proxy HGs (Table S15).”. Otherwise, the whole paragraph makes no sense.

- Page 13: In the paragraph about ME contribution in local NE, I would modify the following sentence: “Moreover, since according to the archaeological data the early NE farmers reached EIAIp by ~5100 BC, the estimated time of admixture between ME and NE groups (~from 6100 to 5100 BC) could also support both local and non-local admixture in this alpine region.” or “Moreover, since according to the archaeological data the early NE farmers reached EIAIp by ~5100 BC, the estimated time of admixture between ME and NE groups (~from 6100 to 5100 BC) cannot exclude both local and non-local admixture in this alpine region.”

Reviewer #2

(Remarks to the Author)

In their rebuttal letter, Croze et al. addressed the vast majority of the reviewers’ comments and requests in a satisfactory manner.

However, I am still unable to validate the authors’ justification for using in-solution capture on remains that are both rare and exceptionally well-preserved, particularly the Mesolithic individual (but that may be expanded to all vestiges preserved to levels that would make shotgun deep-sequencing technically feasible and reasonable in terms of costs). This approach raises major ethical concerns, as it limits future research potential. While the research team has obtained permits (not provided but presumably available to the editorial board), legality does not equate to ethical acceptability. Both researchers and editors must honestly and critically assess whether this protocol aligns with their professional and ethical standards. As a reviewer, I cannot endorse this study unless the authors provide assurances that further deep sequencing remains feasible without depleting the resources (sorry for this term not really appropriate for human vestiges), and that additional sequencing would be approved by the curators and legal authorities in charge of them.

Below, I outline my specific concerns:

The authors themselves acknowledge the importance of one of their individual, whose remains should not be destructively sampled without great care:

“The Mesolithic individual ME (MAD01) is the only Castelnuovian individual known to date on the European continent, and the individual LOS01 is the earliest individual with the Steppe-ancestry component in Italy, bringing great new data to the big picture of European dynamics.”

Given this significance, the decision to apply a targeted capture approach instead of whole-genome sequencing requires a strong justification, which I find lacking.

The authors justify their protocol choices by several arguments:

i) “economic constraints”

The authors cite financial limitations as a key reason for their methodological choices. However, my rough calculations based on the available screening data for MAD01 (20.5% endogenous content, 76 bp average insert size) and sequencing costs at private European facilities indicate that achieving 15X genome coverage (as the authors suggest) would cost under €4,000. This is a modest expense relative to the overall study budget and, notably, lower than the article processing charge for Nature Communications. (Aiming at 1X for many of the other individuals, with endogenous content often higher than 20%, would not be very expensive either).

If funding constraints were an issue, postponing destructive sampling to secure adequate resources should have been

considered. The irreversible consumption of a finite, valuable and irreplaceable archaeo-anthropological resource cannot be justified solely on economic grounds.

A certain amount of the vestiges has been used, this unfortunately cannot be undone. The authors state themselves that they will implement human genomic data for a later project, therefore acknowledging themselves that genomic data will provide additional information. It would be crucial to know if further sequencing of the individuals with decent human DNA content would still be possible. Therefore, the authors should clarify:

-The exact amount of bone/tooth material used per individual, alongside before-and-after sampling images (pictures or scanning).

-Whether any DNA extracts or pre-capture sequencing libraries remain, in sufficient quantities for potential WGS.

-If additional destructive sampling would be permitted by legal authorities, as many curators are justifiably reluctant to approve re-sampling previously analyzed individuals.

ii) the capture approach was adequate to answer the scientific questions.

While capture may address the immediate scientific objectives of the authors, researchers working with ancient human (or any other species for that matter) also have a responsibility to generate reusable datasets. The role of the scientist is not only to answer a specific scientific question but also to build datasets that would be reusable for future studies. This is particularly sensitive when performing destructive analyses of ancient human vestiges. The balance is not only “between the resources available for the project and the informative value of the genomic data we generated”, as it could be for modern genomics for instance. It has to take into account the patrimonial value of these vestiges. The balance would be more between conservation and scientific benefit.

Destructive sampling must balance conservation with scientific value, not just project-specific constraints. When preservation allows, WGS—or at least whole-genome capture—should be prioritized. In any case, it would still be possible to genome sequence the individuals with a relatively high endogenous content and capture the others.

iii) “The capture approach is commonly used in the field of ancient DNA”

The widespread use of capture does not automatically make it appropriate for rare and well-preserved remains (ie outside the “routine” of the field). While I acknowledge that SNP capture is sometimes justified (e.g., for individuals with poor molecular preservation or where similar comparative samples exist), this is not the case here in my opinion.

iv) “we disagree that our data may be less reusable for future studies”

The data generated in this study seem of good quality for capture data, yet they are still lacunary by construct, and therefore less reusable, as the authors state themselves by pointing they want to implement human genomic data in a later project:

“We considered covering this aspect and implementing human genomic data (if possible close to the Iceman genome coverage) for a later project, as soon as we will have the financial support to make it happen. “

This statement contradicts their claim that capture alone is sufficient and suggests that deeper sequencing should have been pursued.

I recognize that Nature Communications may have a different perspective on this issue. However, from an ethical standpoint, I cannot support the publication of a study that conducts destructive analyses on unique and irreplaceable human vestiges without maximizing data recovery. The authors have not convinced me that their capture approach is the most responsible strategy, particularly for MAD01.

Bolzano, 12.20.24

Dear Editor and Reviewers,

We would like to thank you for the opportunity to submit the review of our article and the valuable advice that we have followed scrupulously. Please find below our answers to the Reviewers (marked in blue; the pages refer to the MS numbering displayed in track changes mode).

Yours sincerely,

Valentina Coia, PhD – on behalf of all co-authors

Reviewer's Comments:

Reviewer #1 (Remarks to the Author)

Manuscript #NCOMMS-24-36149, Croze et al., Genomic diversity and structure of prehistoric individuals from the Eastern Italian Alps: Insights from the Tyrolean Iceman's territory.

This manuscript from Croze and collaborators entitled “Genomic diversity and structure of prehistoric individuals from the Eastern Italian Alps: Insights from the Tyrolean Iceman's territory” provides wide-genome data for 47 new individuals from South Tyrol, and 34 new radiocarbon dates, from the Late Mesolithic to the Early Bronze Age. The Mesolithic individual is the only Castelnovian individual known to date on the European continent, and the individual LOS01 is the earliest individual with the Steppe-ancestry component in Italy, bringing great new data to the big picture of European dynamics.

I fully support the publication of this manuscript, in which the analyses are sound and robust, and the interpretations and conclusions well supported.

However, I have some comments, major and minor, mainly to help improving the manuscript, see below.

Major comments

- The introduction needs a bit of work for a clearer description of the context in a European perspective (Neolithic diffusion for instance). Some concepts are missing which are used later in the discussion.

The part link to the main genetic turnovers occurred in Europe has been moved from the Discussion (pag. 11, first paragraph) to the Introduction (pag. 2, 1st paragraph) with some additional information regarding HG groups. Moreover, the two main migration routes in Europe during the Neolithic have been added in the introduction as well as more information regarding Neolithization in the EIALp (top of pag 3).

- Page 6 paragraph 2 and P11 para3. Is this because of the absence of the Fournol ancestry in MAD01 that is also absent from Villabruna but not from Oberkassel? Have you tested it? Because if so, you would have a great hit that your Late Mesolithic Castelnovian individual has connection with the East and could be a representant of recent migrations coming from the South-East, that we haven't been able to show so far by lack of individuals in southern Europe. That would be a very interesting lead to explore in your paper. Given the importance of this Mesolithic sample as the only representant of the Castelnovian culture in southern Europe (except for the Uzzo individuals but Sicily has a different and further history), I think this deserves more digging.

We thank the Reviewer for this important suggestion. We tested the model including *Villabruna* and *Fournol* as sources, but the model resulted not significant (p-value = 0.00118) (top of page 6 of the MS and Table S16) confirming our previous results and indicating no connection between MAD01 and ME individuals with *Fournol* ancestry from more northern and western areas (Posth et al. 2023). The discussion related to this new result and the possible connection of our ME individual with the south-eastern area, as suggested by the Reviewer, has been reported in the Discussion (1st paragraph of page 12).

- P7 para1 and P12. You observe a similar amount of HG ancestry through time in your samples from the Neolithic on, with no "resurgence" as observed in central Europe for instance. You say that MAD01 is the best fit, based on qpAdm 2-way models with alternatively MAD01, Drigge and Villabruna, for the pre-Steppe individuals. I am wondering several things here. Given the constant amount of HG ancestry, another plausible scenario would be that the HG ancestry has been carried on by the first migrant farmers from further East and stays stable through time, with no local admixture. I don't think the qpAdm models can rule out this hypothesis, as the p-value doesn't reflect a "better" fit if higher. I think you should try to disentangle this question using f4-tests and see if you obtain different results according to different HG proxies.

We thank the Reviewer once again for this further important advice. We performed f4 statistics with Villabruna and available Mesolithic individuals from eastern and central Europe and from Italy (we also included Uzzo individuals from Sicily in the analyses, even though this region has a peculiar history as also mentioned by Reviewer 1), and the results were reported in the revised Table S15. Most f4 statistics were negative except for Vela_Spila from Croatia and Villabruna but no test was significant (z-score between -2.8 and +1.069). These results indicate that MN alpine individuals have more genetic affinity to local ME (MAD01) than other HGs used for the analyses. Notably, the two tests with the strongest affinities with MAD01 (Zscore <-2.7) were the comparisons with ME from Iron Gates, practically excluding the Balkans as a region where NE individuals could have obtained substantial amounts of their HG ancestry before reaching the EIAI. These new results were added in the MS (Results top of page 7; 3th paragraph of page 13 in the Discussion section).

Also, the admixture date estimate raises questions here. I find it strange that you got no or poor success applying DATES, and I would consider running it with a group of HG, instead of only one individual (cf. Chintalapati et al 2022 <https://doi.org/10.7554/eLife.77625>), this improves the inference. **Precise results on the admixture date would help the discussion about local vs. non-local HG component.**

The sentence written P12 para 2 “Analysis of the material culture suggests a gradual process of acculturation of the ME groups^{13,10} while the presence of a gap in the dating between ME and NE underlines the discontinuity^{72,73}.” would also go in the sense of **no contacts locally**, according to the radiocarbon dates.

If different proxies are used and work for DATES, I would suggest to rerun the qpAdm accordingly as well, to have a consistent set of analysis.

Following this suggestion, admixture dates estimated in our ME (MAD01) were performed by DATES using a group of WHGs (Luxembourg_Loschbourg.DG; Italy_Villabruna; Spain_Labrana; Switzerland_Bichon.SG; Italy_OrienteC) instead of the single individual of Villabruna (results are reported in Table S20) to improve the inference. It was possible to estimate that the admixture event between WHGs and EHG in MAD01 occurred between 13.735 BC and 8.326 BC and by qpAdm analysis we found that the ancestral model (group of WHGs + EHG) is valid ($p=0.0482$) (Table S16).

Therefore, we also performed a new analysis with DATES to estimate the admixture time in our MN, CA, CA_EBA_EBA and STE05 between HGs and NE (Turkey_N) using a group of WHGs (Hungary_Koros; Germany_Blatterhohle; Switzerland_Bichon.SG; Croatia_VelaSpila; Italy_OrienteC; Falkenstein; BerryAuBac; Iboussieres25-1; Iboussieres31-2; Rochedane; Drigge, MAD01) instead of the single individual of MAD01 (Table S20). We used the same HGs individuals used in the last paper related to the Iceman (Wang et al 2023) with the addition of MAD01 and Drigge (as we have shown with qpAdm that they fit the model) (Table S17). As expected, we found an earlier admixture date in the MN group (between 6,073 to 5,117 BC, median 5,595 BC) than for CA (5,646-3,817 BC, median 4,731 BC), CA-EBA (5,157-3,904 BC, 4,531), and STE05 from MBA (4,648-2,711 BC, median 3,679 BC). Moreover, we found aberrant results for the EBA group (as in the first analysis with MAD01 + AnatoliaNE) (Table S20).

Additionally, the model (WHGs and AnatoliaN) was tested in qpAdm (Table S19). We found that the model is fitting the data for all alpine groups and MBA individual (MN, CA, EBA, STE05) except for the CA_EBA (p -value=0.002) so the admixture time for this chronological group should be taken with caution.

In summary, by combining both analyses (DATES and qpAdm), we reported the estimated admixture time for MAD01. In addition, in agreement with but with greater accuracy than the original analysis, we also reported estimates for the MN (and other groups). These new results were reported in more detail in the revised SM (pages 37-38) file and some of them were discussed in the main text (1st paragraph page 6 and end of page 7 in the Results section).

Combining all results we concluded that (page X of the MS): “Overall, our study suggests (f4, qpAdm) a genetic contribution of local HGs in alpine MN farmers while there may have been some contribution from other HG groups before reaching the EIALp, but not from the

Balkan area. Moreover, since according to the archaeological data the early NE farmers reached EIAp by ~5100 BC, the estimated time of admixture between ME and NE groups (~from 6100 to 5100 BC) also suggests both local and non-local admixture in this alpine region”.

- Also, why didn't you apply an IBD sharing approach to your analysis, for instance with ancIBD (Ringbauer et al 2023, <https://www.nature.com/articles/s41588-023-01582-w>)? You have such a good coverage overall that it would be very interesting to explore the relationships between your groups and the neighboring groups through time, to explore more in depth the geographic networks.

This would also be very interesting for the Neolithic individuals, to disentangle the links with the North vs. the South of the Alps, even if the number of samples and their coverage in the South would be limiting.

We agree with the Reviewer that this would be interesting to test. However, this analysis will be considered in the future using higher resolution data on alpine prehistoric samples and more data from Neolithic individuals from Italy and southern Europe.

Minor comments

Abstract

52 individuals are mentioned in the abstract, but only 47 are used in the study. I would make it clear from the abstract, to avoid any kind of “misleading advertising”.

I would precise which HG groups you talk about for the admixture of the Mesolithic genome. The differences of the Iceman being exclusively linked to uniparental haplogroups, I would make this clear here. Otherwise, it is unclear that you use his genome to compare the others, speak about “continuity” and then state he is different...

We follow the specific suggestions made by the Reviewer. Additionally, taking into account the limit of the words and the new results after additional requested analyses, the abstract was changed substantially. For example, the sentence related to the Iceman and the unilinear transmitted markers was removed to give some space to more remarkable results such as the finding about local and non-local admixture between Mesolithic and migrant Anatolian farmers.

Introduction

- Overall, double check the calls to the references, some commas are missing, and the numbers are unclear (e.g. 34 instead of 3,4). Also, double check the reference list, some of them are wrongly formatted (e.g. 45, 77, 88...).

Some references have been changed following the revision of the text, but all have been carefully checked.

- P3 para1. You mention the Trentino-Alto Adige region from the first sentence, but don't state why it's important for the reader to know that. Specify that's the target of your study.

We added that the Trentino-Alto Adige is the region under study (introduction, first line).

- P3 para2. You don't write "Neolithic" in the first sentence. You have to properly define it before using the term afterwards.

The term "Neolithic (NE)" has been added in the text (end of page 2).

- P3 para2. Name the "Impressa" group with its name. Overall, we miss the big picture here: for the Neolithic, we need to know that two currents are present at the European level, to fit in with the local data.

The description of the two main migration routes in Europe during the Neolithic have been added in the introduction as well as more information regarding Neolithization in the EIAIp (top of pag 3). The correct name has been added (Impressed Ware culture).

- P3 para2. "From the mid-4th mill. BC, within the CA (3500-2200 BC), significant social changes occurred in the EIAIp with the formation of cultural groups distinguished mainly by different funerary practices": which ones? What is different?

The specifics of the funerary practices and the differences have been better explained in the introduction (2nd paragraph of page 3).

Results/Discussion

- P5 para3. Define "1240K" the first time you mention "1.24 million SNPs".

1240 K SNPs has been better defined in the text (top of page 5).

- P5 para3. When you mention the related individuals, mention how many of them (or how many pairs).

The total number of individuals (16) which resulted related based on full consistency between at least two of the applied methods were added in the new version of the manuscript (end of first paragraph, page 5). Moreover, in the same paragraph, in order to define the Middle Neolithic (MN) analyzed in this study and make our dataset clearer, another sentence was added with more details on the total number of individuals from different chronologies: "...which comprises: one Late ME, eight Middle NE (MN), sixteen CA, sixteen CA/EBA and nine from the EBA to the MBA (Table S1 and Table S4)".

- P6 para1. In the first sentence, maybe add “younger” when you talk about “all other alpine samples”, or “as expected for a Mesolithic individual”: it’s so obvious that it took me a while to understand why you were saying this...

We have included the “most recent alpine individuals” (page 5).

- P6 para4. You never explain why you don’t have local EN (or earlier Neolithic) samples. Are the remains missing in the region? Because in the introduction you state that the Neo starts around 5100BC, but your earlier individual is from 4600BC.

Indeed, Early Neolithic specimens have never recovered in this region to date. We reported a short sentence in the original MS (Discussion, top page 13): “...We can assume an even lower contribution by HGs among the first farmers, never recovered before in this area..”. However, we agree that this information was a bit hidden and not clear. The entire paragraph was reformulated and we decided to remove the sentence on HGs contribution since we cannot verify it (top of page 13). Moreover, in order to better clarified that we analyzed individuals for the Middle Neolithic (this is the reason why we have the oldest one of approx. 4600 BC), we added this information in the paragraph “Alpine dataset” in the revised version of the MS, as mentioned before (page 5).

- P8 para2. In the first sentence, why “may explain”? What other explanation would it be for the position of this individual on the PCA? I think you can use “does” or directly “explains”.

The word “may” was removed (page 8).

- P7-8. In the whole section about CA-EBA, I would mention here that you have tested the relatedness between your samples and the published ones and didn’t find any close one. You explain it in the SI, but I would add this here.

In the main text we reported that overall, the f4 statistics do not show clear genetic similarities between Alpine CA_EBAs and specific groups (Results, 1st paragraph of page 8,) and we refer to the MS for more details (page 36).

- P12-13. I’m surprised not to find any discussion about the YC haplogroup G2a, as a marker of affinity between the Alpine individuals and the Balkans or more Northern regions for the Neolithic.

f3 and f4-statistics on autosomal data did not reveal specific affinities between Alpine individuals and those from NE of different origin (Central and west Europe, Balkans). However, we agree that from a paternal perspective and due to the distribution of G2a* in ancient samples analysed so far, G2a* can be considered as a marker of migrations from south-easter and more norther areas. Few sentences were added to the main MS (end of page 14 and top of page 15).

- P13 para3. Why is this paragraph here? It is about Neolithic, I don't understand why it is not with the discussion about Neolithic P12, it's confusing to me and make the discussion harder to follow.

The original idea was to highlight this exception in the context of the other alpine males in the section of the discussion specifically dedicated to the Y-Chromosome. However, we agree with the Reviewer that this may confuse the reader. Then, this part was removed from the original location (discussion, page 14) and was partly moved to the results section (top of page 10) and in the SM (Supplementary Text S1, page 7).

- P15 para3. "Moreover, from an archeological perspective, the appearance of the Steppe-related ancestry in the EIALp is rather unexpected, as there is currently no clear evidence of cultural contacts with groups from the Steppe region." This is extremely simplistic... I don't believe your intention is to mean that actual Steppe groups came into direct contact with the EIALp, and I think this needs to be rephrased.

We apologise for the inattention and thank you for the notification. No, that was not our intention, so the sentence has been reformulated (half of page 16).

- P15-16. At the end of the page 15, beginning of 16, given the number of available individuals, I would tone down a bit the interpretation of kinship practices and population structure.

We agree that this part was a bit speculative. Therefore, and taking into account the suggestions of the Reviewer 2, the sentences related to the discussion on kinships at Y-chromosome and mtDNA level has been eliminated (half of page 17).

- P16. How do you explain the derived allele at position rs1495741 in the gene NAT2 for the Mesolithic individual? There is no discussion about that, though it seems to be an unexpected result. Please develop.

Our results are consistent with what has been observed in the study from Mathieson et al. (2018) that shows no change in the frequency of derived allele of the tag SNP rs1495741 over the past 10,000 years, from hunter-gatherers to groups with Steppe-related ancestry (Mathieson et al. 2018, Fig. 5a). Few sentences related to the comparison with the mentioned study have been added in the discussion (top of page 17). However, the significance of this result is not clear to us and is not even discussed in detail in the cited article. In addition, at the end of the results section (and in the SM), we have made few changes to better clarify the right references associated with SNP rs1495741 and the presumed advantage in agricultural populations (end of Results section of page 11 and end of page 17 in the Discussion section).

Material/Methods

- P17. In the section “Molecular analyses and Radiocarbon dating (14C)”, you mention that the samples were enriched for “more than 2 million polymorphic sites in the human genome using the in-solution target capture kit myBaits[®] Expert Human Affinities – Prime Plus (Arbor Bioscience)”. What are these 2 million SNPs? You didn’t use the kit “ancestral” (or I don’t remember the exact name, as it is not commercialized anymore) which would have had more SNPs, but keep using the 1240k SNPset in your analysis. I’m confused here.

We apologise for the inaccuracy, and we confirm that we didn’t used the “ancestral” kit. The sentence has been corrected (end of page 18).

- P17. In the section “Molecular analyses and Radiocarbon dating (14C)”, you mention 37 minus 2 radiocarbon dated individuals, so 35, when you say 34 in the introduction. Please doublecheck your numbers.

Thank you for the tip; the number 37 was replaced with 36 (top of page 19).

- P17. About the contamination + quality control, I would tell the number of individuals excluded from the analysis in the main text.

The requested information has been included in the revised MS (first paragraph, page 5).

- P19. In the section “Datasets”, the minimum threshold used for the analysis is 5,000 SNPs, that’s very low. I don’t know if there is a consensus about that, but usually the limit is set at 20,000 SNPs on the 1240K dataset... Can you justify why you went that low? Didn’t it affect the PCA? The analysis? According to my experience, I think it does...

We apologise for this error and oversight, as our threshold for the final dataset was actually 20,000 SNPs (from 20021 to 593118 SNPs). We have provided the number of SNPs used for the analysis in Table S10 (see new columns J and K) and correct the main text (page 21).

- P20. In the section about qpAdm, you don’t mention the four-way model used in the main text and in Figure 4 for ROM402.

The description of the four-way model has been added to the Materials and Methods section of the main MS (page 23).

Also, why do you set the p-value threshold at 0.01, and not 0.05 like commonly used? Then of course, most of your models in qpAdm work...

We agree that the threshold at 0.05 is commonly used but all cutoffs are arbitrary, especially for p-values. However, all values are given in the supplementary tables so that the readers can also interpret them according to different cutoffs if desired.

In the study, we tested many models (e.g. Table S17), and a cutoff of 0.05 would have rejected one in 20 even if they were true. The lower cutoff allows us to discuss them at

more resolution and there are some plausible models with $0.01 < p < 0.05$. The multiple testing issue is also illustrated by MOL01 or ROM402 where otherwise none of the models would have been fitting for some individuals (even the plausible model).

- P21. For DATES, see my comment in the results section.

- Tables S2 and S3: I found the two columns “# 1240K Sites (All)” and “# 1240K SNPs (All)” very confusing by their labels. Why do you need the second one, when this label is usually used for what you mean here in the first one? I would suggest simplifying, or making it clearer what you mean in the column headers maybe?

The column “1240K SNPs (All)” has been removed from the Tables, as suggested

- Text S6. The discussion here about the two currents of Neolithic diffusion would have a better place in the main text.

Please refer to the answers above

- Figure S29. There is an error in the FALSE/TRUE colors in the 4th plot on the first line.

We apologise for the oversight. The incorrect Figure S29 has been replaced with the correct one in the revised SM file.

Reviewer #2 (Remarks to the Author)

Croze et al generated and analyzed new ancient DNA data from 47 individuals from the Eastern Italian Alps (also referred to as “ the Tyrolean Iceman’s territory”), dating from the Mesolithic to the Middle Bronze Age. The main objective is to investigate how the genomic structure varied over time in this region, and incidentally bring more context to the famous Iceman. They also report 34 new C14 dates.

Croze et al first screened the NGS data obtained from ancient DNA extracts to assess DNA preservation in the vestiges, then performed in-solution target enrichment for the libraries with >1% human DNA content, followed by DNA sequencing. They investigated migrations events, admixture between local hunter-gatherers and incoming farmers, timing of the arrival of people with Steepe-related ancestry, biological relatedness and a few phenotypic traits.

The main conclusion is a relative stability and isolation of the prehistoric alpine.

The analysis performed are all in line with what is used “in routine” for human ancient DNA studies, but applied here to a new original dataset. No new type of analysis were performed, including for phenotyping for instance, where the few phenotypic traits examined have already been explored in previous studies.

It somehow reads like the authors do not have a strong research question, apart from filling a gap in the ancient human datasets (indeed, few ancient human DNA have been sequenced in this region). They do not developed any new method, nor proposed any

improvement in previous methods.

My main concern is about the methodology followed to generate the data. The authors have access to a large and interesting set of human vestiges. Following screening, it appeared that many extracts are very rich in human DNA (up to 75%!), including the rare individual from the mesolithic (20%). The choice of performing capture in such a case, and not doing shotgun sequencing of whole genomes is extremely questionable. This is even more questionable when the authors are aware of the biases introduced by such capture, and aware of how exceptional the remains they had access to are (page 11: “little is known about the last phase of ME in Europe...MAD01 is one of the rare late ME samples from Italy and Southern Europe genomically analyzed to date... making the data from our alpine ME sample particularly valuable”, page 12: “These are the first genomic data available on NE samples from Italy”, page 13 “ISE01 represents the only burial found in northern Italy attributable to this cultural context”). The authors themselves maintain some confusion between “genome” (term usually used for whole genome sequencing at at least 1x), and what they call “genomic data” (which is actually a set of captured SNPs). By performing capture, the authors lost a vast amount of very valuable genomic information (especially for the rare mesolithic individual), and also metagenomic data (especially for DNA extracted from teeth), that could have been extensively re-used in future studies.

The decision to follow the capture approach rather than shotgun sequencing of whole genomes is based on several reasons, mainly economic and scientific. These reasons are given in more detail below:

- i) economic constraints, as the cost for genomic enrichment is still much less expensive than the whole genome with a high coverage that can justify this analysis and, as usual, we have to deal with the resources available for the study;
- ii) the amount of genomic data generated with the capture approach we expected from our samples was adequate to answer our scientific questions. In addition, it was very important for us to include all individuals in the analyses and not rely on the best ones (highest HR) with a possibly better coverage. Although we generally agree with Reviewer 2 that more data would be desirable, we had to find a balance between the resources available for the project and the informative value of the genomic data we generated. We therefore decided to choose the enrichment approach to obtain a broader alpine dataset.,
- iii) as also acknowledged by Reviewer 2, the methodology we used to generate the genetic data is in line with what is “routinely” used in the field of ancient DNA and is based on well-established method. The capture approach is commonly used in the field of ancient DNA as evidenced by several very recent articles (e.g. Olalde et al. 2023, *Cell*; Nakatsuka et al. 2023, *Nature*). It provides a lot of information that can be compared with large datasets, such as the AADR v54 dataset (Mallick et al. 2024) on 1240K SNPs, which were also generated by the enrichment method. Moreover, at these specific genome positions, good

coverage is required for the analyses, which cannot be guaranteed by shotgun sequencing, resulting in the loss of several informative positions. This makes some comparative analyses difficult due to the lack of overlapping data;

iv) we believe that if one decides to use whole-genome data, one should aim for high coverage data (e.g. Iceman 15X by Wang et al. 2023). Indeed, there are clear limitations associated with low-coverage whole-genome data, as mentioned above. Moreover, despite the economic efforts, the shotgun sequencing of whole genome can produce very different coverages within the same dataset, with only a few samples with relatively 'high' coverage (e.g. Arzelier et al. 2024), preventing a proper comparison between analysed samples.

Furthermore, we disagree that our data may be less reusable for future studies, as we provide good quality data with a great scientific contribution in the field, as state by Reviewer 1 (...”bringing great new data to the big picture of European dynamics....”). Regarding potential bias using the Arbor kit, we took all precautionary measures and proved that our results and findings are authentic and not affected by bias. (Supplementary Text S8).

Finally, the objective of our study concerns human population genetics with clear objectives as reported in the last paragraph of the introduction. Thus, metagenomic analysis was outside the scope of this study. We considered covering this aspect and implementing human genomic data (if possible close to the Iceman genome coverage) for a later project, as soon as we will have the financial support to make it happen.

It is not clear how the libraries were built. The reference for library protocol dates from 2010. Is it exactly this one that was used? If I am correct, this protocol is single-indexed. Then how did the authors ensure their data did not suffer from index-hopping.

Double-indexed libraries were construct followed the protocol from Kircher et al. 2011 (<https://academic.oup.com/nar/article/40/1/e3/1287690>), but we had missed this reference in the text. This has now been added to the revised version of the MS (citation 103 and page 18).

The vast majority of the conclusions are presented in a cautious way and are supported by the data. Some would need to be better justified, or toned down.

For instance:

- Page 15: “the origin of SIU01 outside the territory of EIALp” Only isotopic data could help answering this

We think this is likely but agree with Reviewer 2 that this could only be confirmed by additional analyses. Then, the sentence has been removed (end of page 16).

- Page 15: “we suggest that the observed genetic pattern could be explained by patrilocality” . The data cannot support this hypothesis. All males generated have the

same Y haplogroup, therefore it is not possible to observe, with this type of data, if males were incoming from another settlement from this region. In addition, as the authors state themselves page 16 “ most of analysed males resulted as not being closely related”

We agree that this alternative hypothesis cannot be excluded. Hence, we removed “we suggest” and we add the sentence: “However, with our data, it cannot be excluded that the males analysed came from another settlement in this Alpine region” (page 16). Moreover, the general interpretation of patrilocality was toned down also taking into account the other point raised by the Reviewer 2 (below) and Reviewer 1.

The sentence page 15 “ the union between second degree cousins... was found to involve especially males (3 out of 4)” does not make sense to me. Said males are the offspring, they are not involved in the union.

We apologize for the inaccuracy. Actually, we meant that individuals who turn out to be offspring of second cousins are almost always male. However, taking into account the other comments on the interpretation of patrilocality, this sentence has been removed (half of page 17).

About the archaeological description in the supplementary data: It is sometimes difficult for the reader to pull out the information that is useful for understanding the paper conclusions. In this section, we suggest the authors specify the location where the vestiges are currently stored, and which authorization (from which legal authority, permit number...) they received to perform destructive analysis.

In the introductory part of the supplementary data (Text S1, Archaeological context, page 2), general information is provided about the location where the human remains were found, as well as who conducted the excavations and where the remains are stored. For further clarification, we have added a sentence in the same paragraph (Text S1, end of 1st paragraph page 2), specifying the institutions that authorized the study, which are also listed in more detail in the main manuscript (page 18). Additionally, in the Supplementary data, at the end of each paragraph that describes what is known about the burial sites, a summary of the individuals (if more than one) actually studied is provided, so that the specific details of each individual can be extracted.

It is unclear how the individuals have been classified as EBA; EBA/MBA or MBA. Is this classification based on archaeology or C14 dating? In particular, STE01 is EBA, STE03 is EBA/MBA and STE05 is MBA, while the 3 have overlapping C14 dates, STE03 and STE05 are from the same burial, and the 3 of them are related. This should be clarified, especially as the supplement states it is MBA, with a different date.

The individuals' chronologies are based on the 14C results. The grave goods published by Perini (2001) found in the Stenico tumuli are attributed to Fivè 5b and 6 (Late Middle Bronze Age phase, please see the Supplementary Data at page 25). The new dates from

this study lead to a revision of the materials, which is currently underway, to check for the presence of older archaeological specimens.

About the determination of the individual age: in the supplementary tables, when the authors state “revised for the present study”, please specify which method was used to determine the age at death.

The methodological approach applied during the data revision phase is now outlined in both Table S1 and S2, as follows: ° Anthropological data from Paladin 2011/2012 revised for the current study through standard methods, such as: dental development and eruption patterns (AlQahtani S.J., Hector M.P., Liversidge H.M. Brief communication: The London atlas of human tooth development and eruption. *Am. J. Phys. Anthropol.* 142, 481-490. 2010), measurements of the maximum diaphyseal lengths and the assessment of the epiphyseal–diaphyseal fusion (Schaefer M., Black S., Scheuer L. *Developmental juvenile osteology: A Laboratory and Field Manual.* Elsevier Academic Press, London 2009).

As a general note, we should try to refrain from using the term “sample” when writing about vestiges from deceased individuals, as some people may find it offensive and disrespectful.

Except when referring to methods and DNA samples, in the text ‘sample(s)’ has been replaced by ‘individual(s)’.

Additional changes in the MS made by the authors:

- Following the GUIDE TO FORMATTING ARTICLES, the original order of some sections was changed accordingly (e.g. Acknowledgments, Author contribution and Competing Interest Statement have been moved after References) (not shown).
- One paragraph with the description of authorizations at the study, has been included at the beginning of material and methods section (page 20; Authorizations by local Authorities).
- Abstract: taking into account the limit of the words and the new results after additional requested analyses, the abstract was changed substantially. Please also refer to the abstract of the first submission to see the changes.

Dear Editor,

We would like to thank you for the opportunity to submit the last revision of our article.

Please find below our answers (in blue) to the Reviewers. Furthermore, we revised our article following all the editorial requirements listed in the author's checklist attached to the revision. Finally, as suggested, we have added a sentence (at the end of the Discussion) to meet the Rev2 requirements as indicated with more detail in the author checklist file.

Yours sincerely,

Valentina Coia, PhD – on behalf of all co-authors

REVIEWERS' COMMENTS

Reviewer #1 (Remarks to the Author):

Manuscript #NCOMMS-24-36149A, Croze et al., Genomic diversity and structure of prehistoric individuals from the Eastern Italian Alps: Insights from the Tyrolean Iceman's territory.

I want to thank the authors for their efforts answering my comments, and for the extra analysis provided.

I have a few additional comments before fully validating the manuscript for publication.

- Typos are still present, such as "LBK" at the end of a paragraph for no reason, page 3. Please double check your manuscript.

Thank you for the tip. The manuscript was checked in detail.

- Page 7: I appreciate the effort concerning the additional f4-tests, but I think a rephrasing is needed here. You cannot talk about "negative" results if you don't provide the structure of the f4 test. Here you need to write "..., an f4 statistical analysis on the form f4(HG_test, MAD01; MN, Mbuti) was performed with different sources of proxy HGs (Table S15).". Otherwise, the whole paragraph makes no sense.

The sentence: "... an f4 statistical analysis on the form f4(HG_test, MAD01; MN, Mbuti) was performed with different sources of proxy HGs (Supplementary Data 15)..." has been added to the second paragraph of page 7.

- Page 13: In the paragraph about ME contribution in local NE, I would modify the following sentence: "Moreover, since according to the archaeological data the early NE farmers reached EIAIp by ~5100 BC, the estimated time of admixture between ME and NE groups (~from 6100 to 5100 BC) could also support both local and non-local admixture in this alpine region." or "Moreover, since according to the archaeological data the early NE farmers reached EIAIp by ~5100 BC, the estimated time of admixture between ME and NE groups (~from 6100 to 5100 BC) cannot exclude both local and non-local admixture in this alpine region."

We slightly modified the sentence as suggested by the Rev1 (third paragraph of page 13). We wrote: “the estimated time of admixture between ME and NE groups (~from 6100 to 5100 BC) could also support both local and non-local admixture in this alpine region”.

Reviewer #2 (Remarks to the Author):

In their rebuttal letter, Croze et al. addressed the vast majority of the reviewers’ comments and requests in a satisfactory manner.

However, I am still unable to validate the authors’ justification for using in-solution capture on remains that are both rare and exceptionally well-preserved, particularly the Mesolithic individual (but that may be expanded to all vestiges preserved to levels that would make shotgun deep-sequencing technically feasible and reasonable in terms of costs). This approach raises major ethical concerns, as it limits future research potential.

While the research team has obtained permits (not provided but presumably available to the editorial board), legality does not equate to ethical acceptability.

Both researchers and editors must honestly and critically assess whether this protocol aligns with their professional and ethical standards.

As a reviewer, I cannot endorse this study unless the authors provide assurances that further deep sequencing remains feasible without depleting the resources (sorry for this term not really appropriate for human vestiges), and that additional sequencing would be approved by the curators and legal authorities in charge of them.

Below, I outline my specific concerns:

The authors themselves acknowledge the importance of one of their individual, whose remains should not be destructively sampled without great care:

“The Mesolithic individual ME (MAD01) is the only Castelnovian individual known to date on the European continent,

and the individual LOS01 is the earliest individual with the Steppe-ancestry component in Italy, bringing great new data to the big picture of European dynamics. “

Given this significance, the decision to apply a targeted capture approach instead of whole-genome sequencing requires a strong justification, which I find lacking.

The authors justify their protocol choices by several arguments:

i) “economic constraints “

The authors cite financial limitations as a key reason for their methodological choices. However, my

rough calculations based on the available screening data for MAD01 (20.5% endogenous content, 76 bp average insert size) and sequencing costs at private European facilities indicate that achieving 15X genome coverage (as the authors suggest) would cost under €4,000. This is a modest expense relative to the overall study budget and, notably, lower than the article processing charge for Nature Communications. (Aiming at 1X for many of the other individuals, with endogenous content often higher than 20%, would not be very expensive either).

If funding constraints were an issue, postponing destructive sampling to secure adequate resources should have been considered. The irreversible consumption of a finite, valuable and irreplaceable archaeo-anthropological resource cannot be justified solely on economic grounds.

A certain amount of the vestiges has been used, this unfortunately cannot be undone. The authors state themselves that they will implement human genomic data for a later project, therefore acknowledging themselves that genomic data will provide additional information.

It would be crucial to know if further sequencing of the individuals with decent human DNA content would still be possible. Therefore, the authors should clarify:

- The exact amount of bone/tooth material used per individual, alongside before-and-after sampling images (pictures or scanning).

- Whether any DNA extracts or pre-capture sequencing libraries remain, in sufficient quantities for potential WGS.

- If additional destructive sampling would be permitted by legal authorities, as many curators are justifiably reluctant to approve re-sampling previously analyzed individuals.

ii) the capture approach was adequate to answer the scientific questions.

While capture may address the immediate scientific objectives of the authors, researchers working with ancient human (or any other species for that matter) also have a responsibility to generate reusable datasets.

The role of the scientist is not only to answer a specific scientific question but also to build datasets that would be reusable for future studies. This is particularly sensitive when performing destructive analyses of ancient human vestiges. The balance is not only “between the resources available for the project and the informative value of the genomic data we generated”, as it could be for modern genomics for instance. It has to take into account the patrimonial value of these vestiges. The balance would be more between conservation and scientific benefit.

Destructive sampling must balance conservation with scientific value, not just project-specific constraints. When preservation allows, WGS—or at least whole-genome capture—should be prioritized.

In any case, it would still be possible to genome sequence the individuals with a relatively high endogenous content and capture the others.

iii) “The capture approach is commonly used in the field of ancient DNA” The widespread use of capture does not automatically make it appropriate for rare and well-preserved remains (ie outside the “routine” of the field). While I acknowledge that SNP capture is sometimes justified (e.g., for individuals with poor molecular preservation or where similar comparative samples exist), this is not the case here in my opinion.

iv) “ we disagree that our data may be less reusable for future studies”

The data generated in this study seem of good quality for capture data, yet they are still lacunary by construct, and therefore less reusable, as the authors state themselves by pointing they want to implement human genomic data in a later project:

“We considered covering this aspect and implementing human genomic data (if possible close to the Iceman genome coverage) for a later project, as soon as we will have the financial support to make it happen. “

This statement contradicts their claim that capture alone is sufficient and suggests that deeper sequencing should have been pursued.

I recognize that Nature Communications may have a different perspective on this issue. However, from an ethical standpoint, I cannot support the publication of a study that conducts destructive analyses on unique and irreplaceable human vestiges without maximizing data recovery. The authors have not convinced me that their capture approach is the most responsible strategy, particularly for MAD01.

In regards on deep shotgun sequences approach and future genetic investigations, we added in the main MS (Discussion section, page 17, line 7) the following sentence: “To further investigate social practices with possible extended kinships, phenotypic traits and additional analyses such as metagenomics in past alpine groups, deep shotgun sequencing of the sampled individuals can be performed with the available biological material”.